# Tumor-associated hematopoietic stem and progenitor cells positively linked to glioblastoma progression

I-Na Lu[1,2,3,11], Celia Dobersalske [1,2,3,11], Laurèl Rauschenbach [1,3,4], Sarah Teuber-Hanselmann[5], Anita Steinbach[1,2,3], Vivien Ullrich[1,2,3], Shruthi Prasad[1,2,3], Tobias Blau[5], Sied Kebir[1,3,6], Jens T. Siveke[2,3,7,8], Jürgen C. Becker[2,3,9], Ulrich Sure[3,4], Martin Glas[1,3,6], Björn Scheffler [1,2,3,10,12] & Igor Cima [1,2,3,12✉]

Brain tumors are typically immunosuppressive and refractory to immunotherapies for reasons that remain poorly understood. The unbiased profiling of immune cell types in the tumor microenvironment may reveal immunologic networks affecting therapy and course of disease. Here we identify and validate the presence of hematopoietic stem and progenitor cells (HSPCs) within glioblastoma tissues. Furthermore, we demonstrate a positive link of tumor-associated HSPCs with malignant and immunosuppressive phenotypes. Compared to the medullary hematopoietic compartment, tumor-associated HSPCs contain a higher fraction of immunophenotypically and transcriptomically immature, CD38- cells, such as hematopoietic stem cells and multipotent progenitors, express genes related to glioblastoma progression and display signatures of active cell cycle phases. When cultured ex vivo, tumor-associated HSPCs form myeloid colonies, suggesting potential in situ myelopoiesis. In experimental models, HSPCs promote tumor cell proliferation, expression of the immune checkpoint PD-L1 and secretion of tumor promoting cytokines such as IL-6, IL-8 and CCL2, indicating concomitant support of both malignancy and immunosuppression. In patients, the amount of tumor-associated HSPCs in tumor tissues is prognostic for patient survival and correlates with immunosuppressive phenotypes. These findings identify an important element in the complex landscape of glioblastoma that may serve as a target for brain tumor immunotherapies.

[1] DKFZ-Division Translational Neurooncology at the West German Cancer Center (WTZ), University Hospital Essen/University of Duisburg-Essen, Essen, Germany. [2] German Cancer Research Center (DKFZ), Heidelberg, Germany. [3] German Cancer Consortium (DKTK), Partner site Essen/ Düsseldorf, Germany. [4] Department of Neurosurgery and Spine Surgery, University Hospital Essen, Essen, Germany. [5] Institute of Neuropathology, University Hospital Essen, Essen, Germany. [6] Division of Clinical Neurooncology, Department of Neurology, University Hospital Essen, Essen, Germany. [7] Bridge Institute of Experimental Tumor Therapy, West German Cancer Center, University Hospital Essen, Essen, Germany. [8] DKFZ-Division of Solid Tumor Translational Oncology, West German Cancer Center, University Hospital Essen, Essen, Germany. [9] Translational Skin Cancer Research (TSCR), Department of Dermatology, University Medicine Essen, Essen, Germany. [10] Center of Medical Biotechnology (ZMB), University of Duisburg-Essen, Essen, Germany. [11]These authors contributed equally: I-Na Lu, Celia Dobersalske [12]These authors jointly supervised: Björn Scheffler, Igor Cima. ✉email: i.cima@dkfz.de

Glioblastoma is the most aggressive brain malignancy in adults, lacking effective treatments and leading to death within a median duration of 15–20 months after diagnosis, despite standard combination of surgery, radio- and chemotherapy[1,2]. Cancer immunotherapy, which aims to prime or boost the body's immune system against cancer cells, may improve the clinical course of glioblastoma. Targeting immune checkpoints in advanced malignancies such as melanoma, kidney and lung cancer, achieved impressive therapeutic effects, sparking new hopes for the treatment of brain tumors[3,4]. However, the glioblastoma microenvironment is characteristically immunosuppressive compared to other malignancies, owing to, at least in part, potent immunosuppressive cytokines such as TGF-β and IL-10[5], negative regulators of effector cell functions such as programmed death-ligand 1 (PD-L1), indoleamine 2,3-dioxygenase (IDO) and oncometabolites such as (R)-2-hydroxyglutarate[6,7]. Accordingly, the use of anti-PD-1 antibodies in recurrent glioblastoma failed to prolong patient overall survival[8]. Transfusion of a single dose of chimeric antigen receptor (CAR) T cells targeting EGFRvIII led to adaptive immunosuppression and therapy failure, indicating that the major barrier for immunotherapy of glioblastoma may lie on the inhibitory tumor microenvironment[9]. Nevertheless, Brown and colleagues observed sustained clinical response for 7.5 months in a patient with highly aggressive recurrent glioblastoma, after application of CAR T cells targeting interleukin-13 receptor alpha 2 (IL13Rα2)[10]. Further, a preliminary report on a phase III clinical trial of dendritic cell vaccine in glioblastoma patients reported a median overall survival of 23.1 months, compared to the 15–20 months achieved with the current standard of care[11] These studies remarkably document the challenging endeavors of immunotherapy in the treatment of glioblastoma.

The brain tumor immunosuppressive microenvironment is marked by the presence of several immune cell types including regulatory T cells and myeloid cells such as bone marrow-derived macrophages lacking T cell co-stimulatory molecules[5]. Glioblastoma-associated immune cells may not only create an immunosuppressive microenvironment but also directly promote malignancy[3,12]. For example, tumor-infiltrating neutrophils facilitated cancer stem cell accumulation through S100A4[13]. Despite efforts in decoding the complexity of the immune system's modus operandi during brain tumor progression, interactions between different cell types in glioblastoma are not yet fully understood. Moreover, knowledge aimed at modulating the immune system therapeutically and in a patient-specific setting is lacking. The systematic, discovery-driven screening of immune cell types in glioblastoma may help to uncover important immunologic targets and lead to the discovery of predictors of clinical outcomes.

Recent studies point in this direction: For example, Gentles et al.[14] profiled the occurrence of 18 distinct immune cell types in various cancer types, revealing unknown links of immune cell types with clinical outcomes. In brain cancers, the relative leukocyte composition was significantly different compared to non-brain solid tumors as exemplified by a decrease in B cell subsets and an increase of monocyte and neutrophil proportions. Furthermore, this report highlighted the discovery of favorable and adverse outcomes for various cell subsets in glioblastoma, demonstrating the benefits of a discovery-driven screening approach in the analysis of the tumor microenvironment.

Here, we have profiled the cellular landscape of brain cancers using a computational approach for transcriptome analysis, separating the signals of 43 different cell types, including 26 distinct immune cell types. We uncover and validate the presence of hematopoietic stem and progenitor cells within brain tumor samples and demonstrate a positive association of this cell population with glioblastoma malignancy and immunosuppression.

## Results

**Estimating the relative abundance of cell types using transcriptomes**. To infer the cellular landscape of brain tumor tissues from transcriptome data, we established Syllogist, a reference-based algorithm for cell type estimation (Fig. 1a, Methods). To this end, we employed a validated gene expression matrix containing cell type-specific transcriptomes[15]. We next extracted data for 43 different cell types, including a selection of 26 immune cell types, similarly to previous studies[16,17]. For each cell type we determined a signature of the top 80 specific genes by calculating specificity indices based on a Shannon entropy-based statistic introduced by Shug et al.[18] (Supplementary Data 1). We next computed the presence of each 80-gene signature in query transcriptomes and compared them with a null model comprising 1000 simulations by Fisher's exact test. The resulting odds ratios were used as proxy for the relative amount of a target cell type to be compared between samples (intersample comparison). The algorithm was validated using a set of previously published positive controls. Each positive control produced specific signals in the corresponding reference samples but not in cells from different ontogenies (Fig. 1b). We also analyzed specific cell types that were not directly represented by our references. For example, freshly isolated glioblastoma cells were specifically assigned to the astrocytic references and neurons to the neuronal lineages. Cancer-associated fibroblasts (CAFs) were distinct from normal fibroblasts and their signals associated with mesenchymal stem cell signatures. Microglia-derived transcriptomes associated with both monocyte and macrophage references (Supplementary Fig. 1).

To test the performance of Syllogist and benchmark it with reported computational methods, we investigated publicly available transcriptome datasets with available paired immunophenotyping data[19]. To this end, we analyzed publicly available PBMC transcriptomes with paired mass cytometry data of 24 immune cell subsets that were previously used for the validation of a similar cell type estimation method[20]. When correlating cell type signals with immunophenotyping data, Syllogist performed similarly to CIBERSORT[21], xCell[20], QuanTIseq[22] and EPIC[23] on 8 commonly detected immune cell subsets (Fig. 1c). In addition, Syllogist performed similarly to all tested methods in estimating CD4 and CD8 T cell subsets in transcriptome data of melanoma and lung cancer tissues paired with quantitative immunofluorescence data[22] (Fig. 1d). To benchmark Syllogist with other established methods for the analysis of brain tissue transcriptomes, we quantified common cell types for all methods and compared the results using correlation matrices for each cell type. This analysis showed that Syllogist was always in agreement with at least two other methods (Fig. 1e).

To specifically interrogate intersample differences in brain tissue cellular composition, we performed 2-sample paired comparisons between 100 brain tissue transcriptomes[24] with and without in silico spike-in transcriptomes for various immune cell types at various ratios (Fig. 1f). A selected number of cell types could be detected at percentages down to 0.05%. For example, brain samples with 0.05% in silico-spiked plasmacytoid dendritic cell (pDC) or naïve CD4 T cell transcriptomes were significantly different from the same brain transcriptomes without spiking. On the other hand, cell types such as naïve B cells could only be detected when spiked at frequencies above 0.8% (Fig. 1f).

These results indicate that Syllogist is able to estimate the relative quantity of 43 distinct cell types from bulk RNA sequencing data. Our algorithm performs similarly to known deconvolution and gene enrichment methods[20–23] in intersample comparisons and it can detect the relative amount of cell types in brain tissue transcriptomes with a limit of detection below 1% when using 2-sample hypothesis testing.

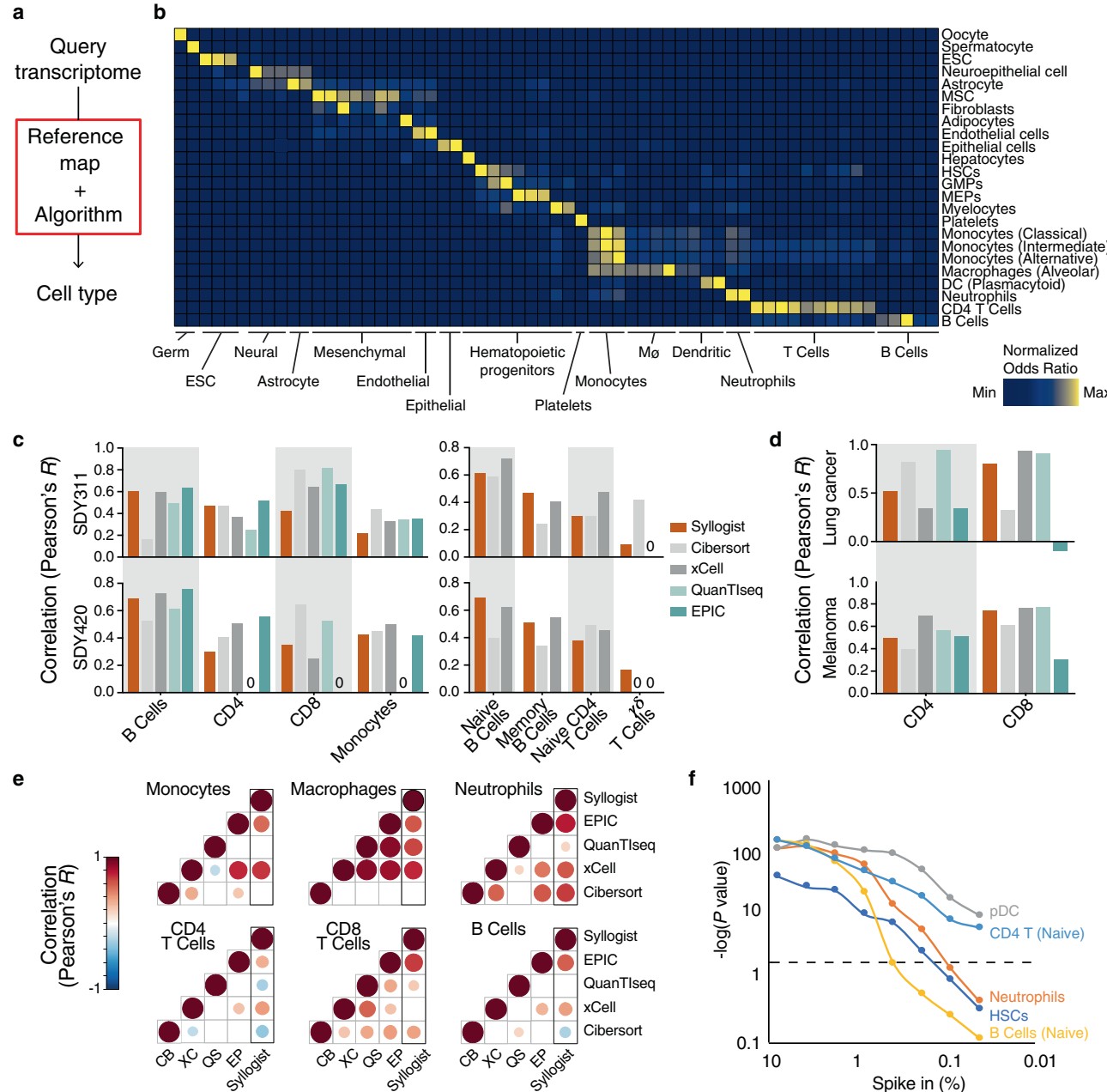

**Fig. 1 Cell type estimation using transcriptomes. a** Simplified workflow of Syllogist for detection and relative quantitation of cell types from bulk tissue transcriptomes. **b** Heat map comparing the number of genes enriched for each published positive control sample (rows) and cell type (columns) over random enrichment. Each colored box represents a normalized odds ratio of the respective Fisher's exact test ranging from 0 (blue) to 1 (yellow). ESC Embryonic stem cell, MSC Mesenchymal stem cell, HSCs Hematopoietic stem cells, GMPs Granulocyte–monocyte progenitors, MEPs Megakaryocyte–erythroid progenitors, DC Dendritic cell. **c** Comparison of Syllogist with CIBERSORT, xCell, QuanTIseq and EPIC. Bar plots represent the Pearson correlation coefficients calculated by comparing the Syllogist odds ratios, the CIBERSORT, xCell, QuanTIseq, and EPIC scores with the quantitative data of PBMC fractions measured by CyTOF (SDY311 [https://www.immport.org/shared/study/SDY311], $n = 61$ patients and SDY420 [https://www.immport.org/shared/study/SDY420], $n = 104$)[19]. **d** same as (**c**), applying melanoma ($n = 32$ samples) and lung cancer ($n = 8$) datasets with paired quantitative data of CD4 and CD8 T cells by immunofluorescence[22]. **e** Correlation matrices represent the agreement of Syllogist with four established computational methods for the indicated immune cell types. Pearson correlation coefficients with $p < 0.05$ are shown as circles, with circle size and color matching Pearson correlation coefficients from −1 (blue) to 1 (dark red). Empty squares represent correlation coefficients of $p \geq 0.05$). CB Cibersort; QS QuanTIseq, XC xCell, EP EPIC. $p$ values determined using a two-tailed Student's $t$-test. **f** Line chart represents −log10 ($p$ values) obtained by comparing paired samples with and without spike-in for the indicated cell types at various percent spike-in ($n = 200$ brain tissue transcriptomes, two-tailed paired Student's $t$-test). pDC plasmacytoid dendritic cells, HSCs Hematopoietic stem cells. The horizontal dashed line indicates the threshold considered for significance ($p = 0.05$). Source data of (**c**), (**d**) and (**f**) are provided as a Source Data file.

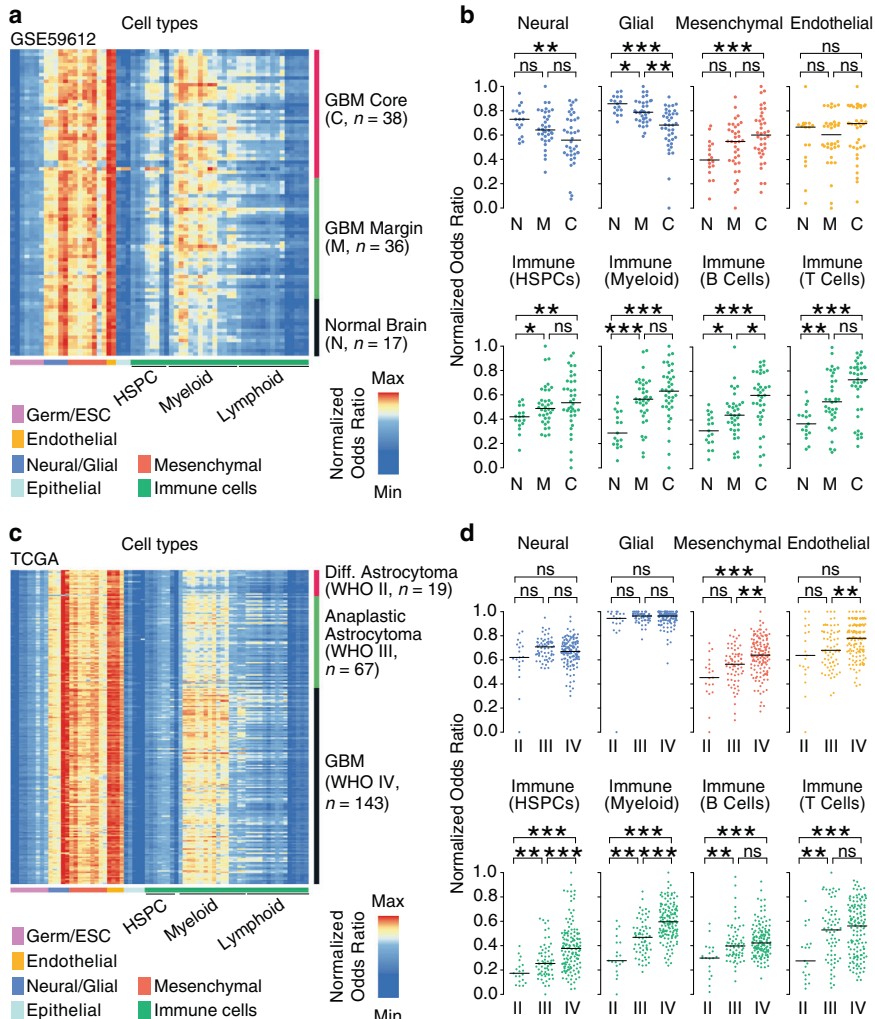

**Fig. 2 HSPCs are enriched in glioblastoma and associate with tumor grade. a** Heatmap represents Syllogist signals for each cell type (columns) and patient samples (rows) for data derived from Gill et al.[26], including core, margin and normal brain tissue samples ($n = 91$ samples). GBM Glioblastoma, HSPC Hematopoietic stem and progenitor cell, ESC Embryonic stem cell, Mø Macrophages. **b** Association of brain sample locations with Syllogist normalized odds ratios of eight selected cellular compartments. Dot plots represent the samples from normal brains (N, $n = 17$ samples), glioblastoma margins (M, $n = 36$), and cores (C, $n = 38$). **c** Heatmap represents Syllogist signals for each cell type (columns) and patient samples for the LGG-glioblastoma cohorts[29,30] ($n = 229$ samples). **d** Association of tumor grade with normalized odds ratios of eight main cellular compartments. Dot plots represent the samples from diffuse astrocytoma (WHO grade II, $n = 19$ samples), anaplastic astrocytoma (WHO grade III, $n = 67$) and glioblastoma (WHO grade IV, $n = 143$). $p$ values were determined by 2-tailed, unpaired Student's $t$-test with correction by the Benjamini–Hochberg procedure. *, $p < 0.05$; **, $p < 0.01$; ***, $p < 0.001$; ns not significant. For (**b**) and (**d**), the exact $p$ values are reported in Supplementary Data 2 and 3, respectively. Source data of (**b**) and (**d**) are provided as a Source Data file.

**The cellular landscape of brain tumors.** Residual tumor cells in glioblastoma remain consistently scattered beyond the surgical margin, facilitating rapid recurrence of disease[25]. The study of glioma cells and their microenvironment at the surgical margin is therefore of utmost clinical relevance as this region represents the target of post-surgical therapies, including immunotherapies. We were therefore interested in profiling the cellular landscapes of glioblastoma cores and their margins. To this end, we used the dataset from Gill *et al.*[26], which includes samples from both tumor centers and margins, as well as normal brain samples. In tumor cores ($n = 38$ samples) and margins ($n = 36$) we observed an increase in myeloid and lymphoid cell types compared to normal brains ($n = 17$), matched by a decrease in neural and glial cell proportions (Fig. 2a, b and Supplementary Data 2). Surprisingly, we also detected increasing signals derived from hematopoietic stem and progenitor cells (HSPCs) in the tumor margins and cores compared to normal brains ($p = 0.018$ and $p = 0.005$,

respectively, Student's $t$-test) indicating that HSPCs might reside in glioblastoma cores and margins (Fig. 2b). For example, among HSPC subsets, we observed an enrichment of hematopoietic stem cells (HSC, $p = 0.024$), granulocyte-monocyte progenitors (GMP, $p = 0.009$), promyelocytes ($p = 0.009$) and myelocytes ($p = 1.46 \times 10^{-5}$) in the glioblastoma margins compared to normal brains. Enrichment of HSPC subsets were also significant in the glioblastoma cores compared to normal brains (HSC, $p = 0.022$, GMP, $p = 0.001$, promyelocytes ($p = 0.0006$) and myelocytes, $p = 4.83 \times 10^{-6}$) (Supplementary Fig. 2a and Supplementary Data 2). Based on these results we hypothesize that HSPCs infiltrate glioblastoma and are enriched not only at the tumor cores but also at their margins. These cells may, therefore, persist in the postsurgical cavity after resection.

The pathology of lower grade isocitrate dehydrogenase (*IDH*) wildtype astrocytoma is difficult to interpret, because *IDH* wildtype diffuse or anaplastic astrocytomas (WHO grade II and

III, respectively) may have clinical courses similar to glioblastoma (WHO grade IV)[27,28]. We were therefore interested in profiling the cellular landscape of these tumors by analyzing the cell type content of the TCGA lower grade glioma (LGG) and glioblastoma cohorts[29,30]. We identified 19 diffuse astrocytomas, 67 anaplastic astrocytomas and 143 glioblastomas, all harboring *IDH1/2* wildtype genotypes. In this data, we detected several differences in the cellular landscapes of astrocytomas between WHO grades (Fig. 2c, d and Supplementary Data 3) and once more, we detected signals derived from various HSPC subsets. In glioblastoma, HSPC signals were significantly higher compared to grade II and III astrocytomas ($p = 2.14 \times 10^{-7}$, and $p = 5.84 \times 10^{-6}$, respectively) (Fig. 2d). In particular, we observed an enrichment of HSCs ($p = 2.82 \times 10^{-7}$), GMPs ($p = 5.68 \times 10^{-7}$), common myeloid progenitors (CMP, $p = 2.34 \times 10^{-5}$), promyelocytes ($p = 2.49 \times 10^{-5}$), myelocytes ($p = 3.55 \times 10^{-6}$) and megakaryocyte–erythroid progenitors (MEP, $p = 0.041$) in glioblastoma compared to grade III tumors. Similar results were obtained when comparing glioblastoma vs grade II tumors (Supplementary Fig. 2b and Supplementary Data 3). A specific subset of myeloid-derived suppressor cells characterized by a phenotype of immature or "early stage" myeloid cells (eMDSCs, Lin$^-$HLA-DR$^-$CD33$^+$)[31] may be responsible, at least in part, for signals detected in the HSPC compartment by Syllogist. However, eMDSCs associated uniquely with the CD14$^+$ monocyte references and not with progenitors of the hematopoietic lineages (Supplementary Fig. 3).

Together, we profiled 43 different cell types in 217 glioblastomas, 86 WHO grade II and III astrocytomas and 17 normal brain tissue samples by gene enrichment analysis. Our results indicate that several HSPC subsets, while expected to reside in the bone marrow or the peripheral blood[32], are detected in brain tumors at their cores and margins. In addition, HSPC signals positively associate with the presence and histological grade of brain tumors.

**HSPCs populate brain tumor tissues**. The detection of endogenous hematopoietic progenitors in human brain tissues represents an intriguing finding which, to our knowledge, has not yet been reported. To test if HSPCs can be identified in glioblastoma tissues by classical immunofluorescence, we determined the status of CD34 and CD45 in paraffin embedded formalin fixed glioblastoma tissues ($n = 4$ patients). In all patients, we detected CD45$^+$CD34$^+$ double positive cells (Fig. 3a). To further study HSPCs in brain tumors, we analyzed a set of 12 fresh surgical tissue samples derived from 12 patients using flow cytometry (7 primary *IDH* wildtype glioblastoma tissues, 4 lower grade gliomas and 1 non-small cell lung cancer brain metastasis, Supplementary Data 4). We interrogated the presence of 7 HSPC subsets[33] in cell suspensions obtained from these tissues and compared them with healthy bone marrow-derived mononuclear cells. All samples stained positive for HSPCs as defined by lineage (Lin) negative and CD34 positive events (Fig. 3b–d). In glioblastoma tissues, we detected a median of 1813 Lin$^-$CD34$^+$ HSPCs per million cells analyzed (range $n = 525$–8882); In lower grade glioma samples, 1617 HSPCs per million ($n = 47$–2707) in the metastatic sample we observed 296 HSPCs per million, compared to 7933 HSPCs per million derived from a healthy donor bone marrow sample (Fig. 3d). Interestingly, we observed a notable lineage bias of HSPC subsets in glioblastoma compared to bone marrow-derived mononuclear cells or lower grade gliomas. In glioblastoma samples we recorded an increase in hematopoietic stem cell proportions (HSCs, Lin$^-$CD34$^+$CD38$^-$CD45RA$^-$CD90$^+$) ranging from 43.0% to 67.5% of total HSPCs compared to 2.6–23.6 % in lower grade gliomas ($p = 4.83 \times 10^{-4}$, two-tailed Student's *t*-test) and

1.9% in the bone marrow sample (Fig. 3d). Multipotent progenitors (MPPs), defined by the expression of Lin$^-$CD34$^+$CD38$^-$CD45RA$^-$CD90$^-$ were also overrepresented in brain tumor tissues compared to the healthy bone marrow sample. These results indicated that immature HSPC subsets, in particular HSCs and MPPs, were enriched in glioma tissue samples and a brain metastasis sample (Fig. 3d). These data were also in line with the detection limit of Syllogist for HSCs reported in Fig. 1f (~1200 cells/million). In addition, our results were not biased by contamination from circulating HSPCs, as the proportion of HSCs in glioblastoma tissue cell suspensions was 4–37.5 fold higher compared to the known proportion of HSCs in the peripheral blood mononuclear cells[34]. Analysis of circulating and tissue-associated HSCs from paired samples confirmed this enrichment ($n = 2$ samples from one patient, Supplementary Fig. 4a). Also, the presence of non-hematopoietic progenitors expressing CD34 such as endothelial and tumor cells[35,36] may have been excluded from our analysis by lineage markers such as CD14[37,38] or CD56[39,40]. In this line, adding an endothelial-specific antibody in our lineage cocktail (anti-CD144) did not change the flow cytometric profiles of HSPCs in glioblastoma cell suspensions (Supplementary Fig. 4b). Similarly, the preferential accumulation of immature HSPC subsets in glioblastoma samples could also be observed when gating for CD45 positive cells to exclude potential contaminants of non-hematopoietic origin (Supplementary Fig. 4c). We next asked if HSPCs could be detected in a publicly available dataset of single cell RNA-Seq from glioblastoma patients[41]. In all samples analyzed, we annotated various HSPC subsets, in particular HSCs and MEPs. Interestingly, in accordance to the flow cytometric profiles, we also noted increased proportions of immature progenitors compared to two healthy bone marrow samples[42] used as positive controls (Supplementary Fig. 4d).

To test the proliferative capacity and lineage commitment of tumor-associated HSPCs, we performed colony-forming cell (CFC) assays using cell suspensions derived from 14 brain tumor surgical specimens cultured in semi-solid media (eight glioblastomas, four lower grade gliomas, and two brain metastases, Supplementary Data 4). We observed hematopoietic colonies in 7/8 glioblastoma patient specimens (median, $n = 6.5$ colonies/sample, range, $n = 0$–13), whereas colonies derived from lower grade gliomas or metastasis could be observed only in 1 ganglioglioma case (median $n = 0$, range $n = 0$–2) (Fig. 3e, f). In glioblastoma-derived cultures we observed a spectrum of CFU-GEMM, CFU-GM and BFU-E colonies, confirming that HSPCs from brain tumor tissues can proliferate and differentiate into myeloid lineages (including erythroid cells). This data supported again our earlier observations consistently indicating the presence of HSPCs in glioblastoma. Furthermore, the colony-forming activity was significantly higher in glioblastoma compared to the non-glioblastoma tumor entities ($p = 0.025$, Fisher's exact test) (Fig. 3f). Moreover, CFU-GEMM colonies, which derive from more primitive HSPCs, were detected exclusively in glioblastoma-derived cultures (Fig. 3f), indicating the presence of more immature HSPC subsets in glioblastoma samples compared to other brain tumor entities. These results were also in agreement with the flow cytometry data reported in Fig. 3d, showing an enrichment of immature hematopoietic progenitors within the glioblastoma microenvironment.

To test if tumor-associated HSPCs displayed similar potency between patients, we selected glioblastoma samples with similar flow cytometric HSPC profiles (Fig. 3d, Patient 3, 8 and 14) and compared their colony formation ex vivo (Fig. 3f). All three patients produced distinct CFC profiles under identical conditions, suggesting heterogeneous potency of tumor associated HSPCs in vivo.

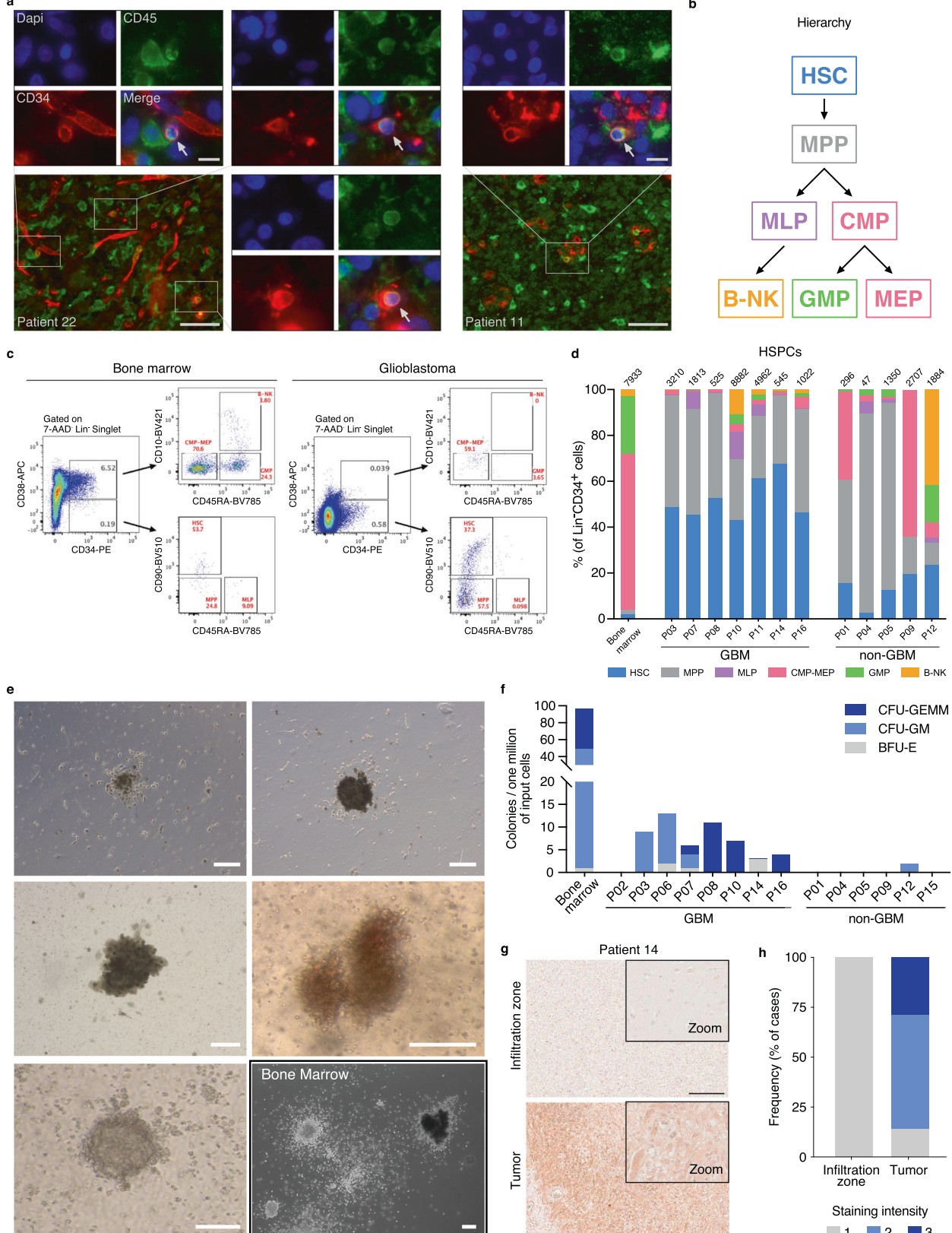

The lineage fate and function of HSPCs in the bone marrow depend on specialized factors such as the CXC chemokine ligand (CXCL) 12, which signals through the CXCR4 receptor to induce HSPC niche colonization, proliferation and differentiation[43]. To test if tumor-associated HSPCs reside in a similar microenvironment, we examined CXCL12 expression in tissue sections from 7

glioblastoma patients, in both the tumor core and the infiltration zone by immunohistochemistry. We observed increased CXCL12 expression in 6/7 glioblastoma tumor cores compared to the peripheral infiltration zone in the same section (Fig. 3g, h). Specifically, CXCL12 was prominently detected in tumor cells with uniform staining patterns within samples. Tumor cells may

**Fig. 3 Characterization of tumor-associated HSPCs in human glioblastoma tissues. a** Representative immunofluorescence appearance of CD34+ (red)/ CD45+ (green) cells (arrows) in formalin-fixed, paraffin-embedded glioblastoma tissue sections from two patients (out of four analyzed, all with similar results). Nuclei were counterstained with DAPI (blue). Scale bars = 50 μm (overview) and 10 μm (insets). **b** Diagram describes the hierarchy of HSPC subsets analyzed in this study by flow cytometry. HSC Hematopoietic stem cell, MPP Multipotent progenitor, MLP Multi-lymphoid progenitor, CMP-MEP Common myeloid progenitor and megakaryocyte-erythroid progenitor, GMP Granulocyte-Monocyte Progenitor, B-NK B-NK progenitor. **c** Representative flow cytometry profiles of human bone marrow and glioblastoma tissue, gated for seven HSPC subsets. Cellular frequencies are highlighted in red. **d** Stacked barplot of seven HSPC subsets observed in glioblastoma (GBM, n = 7), non-glioblastoma tumor tissues (non-GBM, n = 5, Supplementary Data 4) and a healthy donor bone marrow sample by flow cytometry. **e** Representative colony morphologies from CFC assays of glioblastoma cell suspensions derived from a total of eight patients, and from bone marrow-derived mononuclear cells. Scale bars = 100 μm. **f** Barplot indicating number and types of colonies in CFC assays derived from bone marrow, glioblastoma (GBM, n = 8 patients) and non-glioblastoma (non-GBM, n = 6 patients) cell suspensions. CFU Colony forming unit, -GEMM Granulocyte, erythrocyte, monocyte, megakaryocyte, -GM, Granulocyte, monocyte, -E, Erythroid. **g** CXCL12 staining of the infiltration zone (upper panel) and the tumor (lower panel) of the same tissue section. Scale bars = 200 μm. One representative staining of eight shown). **h** Barplot showing percent CXCL12 staining intensity (1 = weakly positive, 2 = moderately positive, 3 = strongly positive) of tumor core and infiltration zones from glioblastoma (n = 16 samples from eight patients). Source data of (**a**), (**d**), (**f**), (**g**), and (**h**) are provided as a Source Data file.

therefore provide the necessary microenvironment for HSPC colonization and multilineage differentiation in glioblastoma.

**Tumor-associated HSPCs show distinct phenotypes compared to bone marrow-derived and circulating HSPCs.** To compare the phenotype of tumor-associated HSPCs with canonical HSPC subsets, we obtained single cell transcriptomes from magnetically enriched CD34+CD45+ *IDH* wt glioblastoma cells (n = 660 cells) derived from a fresh surgical specimen. For comparison, we used transcriptomes derived from bone marrow (n = 549), blood (n = 283) and a fresh tumor-free brain sample (n = 105) that were processed with an identical protocol (Fig. 4a and Supplementary Data 4). In the glioblastoma sample, Uniform Manifold Approximation and Projection (UMAP) and graph-based clustering unraveled 6 clusters (Fig. 4b). Cluster 3 significantly overexpressed the HSPC gene *SPINK2*[44] compared to all other clusters (Supplementary Fig. 5a). Independent annotation using a published reference-based algorithm[45] uncovered the same cluster as containing HSPCs (Fig. 4c). The remaining clusters were annotated as containing mainly myeloid cell types (monocytes, macrophages) or non-immune cells (Fig. 4c). Expression of known progenitor and myeloid markers confirmed our annotation. In particular, the HSPC cluster contained cells expressing *PTPRC* (CD45), cells exclusively expressing *CD34* and hematopoietic progenitor markers *SPINK2* and *GATA2*[44], but lacked expression of myeloid lineage markers *CD14*, *ITGAM* (CD11b), microglia-specific markers (*TMEM119*) or myeloid markers typically expressed by immature (lin-) myeloid-derived suppressor cells (*CD33*) (Fig. 4d). The HSPC cluster also lacked expression of markers specific for lymphoid, endothelial, mesenchymal, astrocytic or neural populations (Supplementary Fig. 5b), confirming the enrichment of our targeted population. In the glioblastoma sample, we annotated a total of 126 tumor-associated HSPC transcriptomes subdivided as follows: HSC (n = 15 transcriptomes), MPP (n = 28), CMP (n = 2), GMP (n = 29), CLP (n = 14) and MEP (n = 38). Notably, in the sample derived from a tumor-free brain region, we failed to detect HSPC-typic transcriptomes, further substantiating a preferential accumulation of HSPCs in tumor tissues. In cells enriched from a healthy bone marrow and blood sample that were used as positive controls, we annotated 500 and 174 HSPCs, respectively (Fig. 4e). We next compared the transcriptomes of tumor-associated HSPCs with our control samples. Comparison was conducted in a normalized, combined dataset to interrogate proliferative states and to determine differentially expressed genes in HSPC subsets between glioblastoma and controls. UMAP plotting of HSPC transcriptomes from glioblastoma, bone marrow and blood clustered within a common region (Fig. 4f). In this topological representation, expression of marker genes for bone marrow-derived

HSPC subsets[46] and our annotated bone marrow sample matched specific regions of the graph. This was mirrored by the same subsets (except for CLP) annotated in the glioblastoma sample, indicating, at least for the HSC, MPP, GMP and MEP subsets, strong similarities between bone marrow and tumor-derived HSPC subsets (Supplementary Fig. 5c, d). Next, for these HSPC subsets, we scored non-cycling or cycling cells using an established algorithm[47]. The proportion of cell cycle phases in HSPCs from the bone marrow sample were in agreement with previously reported data[48,49]. However, we noted a significant increase of cycling MPP in the glioblastoma sample when compared to healthy bone marrow (Fig. 4g). In addition, the number of tumor associated HSPCs with an active cycling profile were proportionally higher compared to differentiated myeloid and lymphoid cells within the same glioblastoma sample (Supplementary Fig. 5e, f). These data suggest that tumor-associated HSPC subsets, in particular MPPs, GMPs and MEPs may proliferate in situ. We next analyzed differential gene expression between HSPC subsets in our combined dataset. We selected genes that were consistently regulated between tumor-associated HSPCs and bone marrow or blood-derived HSPCs (Fig. 4h and Supplementary Data 5–8), for each subset with sufficient cells available. Interestingly, among the top upregulated genes in tumor-associated HSPCs, we noticed genes coding for proteins that were previously shown to impact on hematopoietic stem cell maintenance and cell cycle progression (e.g., *HMGB1*, *SOX4* and *STMN1*, in both HSCs and MPPs[50–52]) or to mediate tumor progression such as *TMSB10*[53]. In conclusion, single cell RNA-Seq analysis of tumor-associated HSPCs confirmed preferential enrichment of these population within glioblastoma compared to normal brain. Moreover, tumor-associated HSPC transcriptomes contained signatures associated with active cell cycle phases and showed enrichment of genes affecting hematopoietic progenitor maintenance and tumor progression, when comparing with healthy bone marrow-derived and circulating HSPCs.

**HSPCs promote a malignant and immunosuppressive phenotype in glioblastoma.** To investigate if hematopoietic progenitors can alter glioblastoma progression and/or immunosuppression, we co-cultured bone marrow-derived HSPCs with three fluorescently labeled glioblastoma cell lines (T98G, LN229, U87) and monitored their proliferation and PD-L1 expression by flow cytometry. After 48 hours, we observed increased proliferation in all three cell lines tested in the presence of HSPCs compared to cultures without HSPCs (Fig. 5a–c). For example, 43.2% of T98G cells underwent at least one cellular division in the presence of HSPCs, compared to 23.4% in the control samples. Besides, we noted a proportion of tumor cells exhibiting accelerated cell cycle progression in the presence of HSPCs (Fig. 5b, d) as 16.7% of

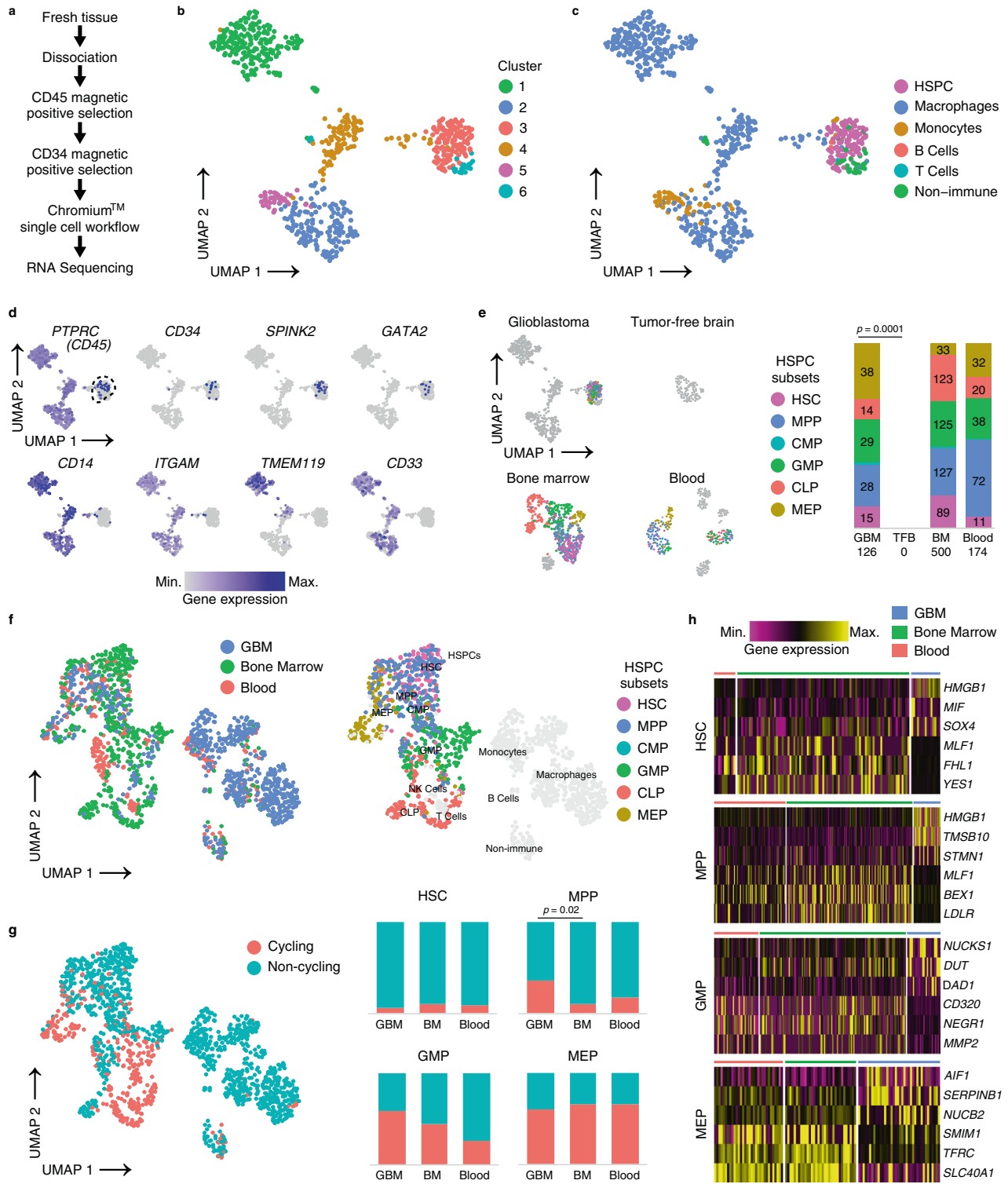

T98G underwent ≥2 division rounds within 48 h, compared to 0% in the absence of HSPCs. In both T98G and U87 co-cultured with HSPCs, we also observed an increase in cell-surface expression of PD-L1 (Fig. 5c), indicating that HSPCs may contribute to the immunosuppressive environment in glioblastoma progression by inducing the expression of immune checkpoint molecules on glioma cells. To test if proliferation and PD-L1 upregulation in tumor cells were caused by a soluble factor or by cell-cell contact, we incubated T98G cells with HSPCs or their conditioned media. The proliferation and PD-L1 expression were induced only in the presence of HSPCs, indicating the requirement for direct cell-cell contact (Fig. 5e). Interestingly, in the presence of HSPCs, PD-L1 expression was upregulated exclusively in the T98G cells that underwent at least 2 divisions but not in the parental cells.

Neural stem cells (NSCs) carrying driver mutations have been proposed as cells of origin for glioblastoma. NSCs has also been showed to preferentially migrate and invade developing gliomas, promoting malignant progression[54,55]. To test if HSPCs may contribute to NSC recruitment in gliomas, we applied an invasion assay using human hippocampal adult human neural progenitors

**Fig. 4 Comparing tumor-associated HSPCs with canonical hematopoietic progenitors by scRNA-Seq. a** Enrichment protocol for derivation of single cell suspensions from biosamples. **b** UMAP projection of CD45+CD34+-enriched glioblastoma cells, color coded for graph-based clusters. **c** Cell type annotation by SingleR in the CD34+CD45+-enriched glioblastoma sample. HSPC Hematopoietic stem and progenitor cell. **d** Marker expression for immune (*PTPRC*), HSPCs (*CD34, SPINK2, GATA2*), and myeloid lineages (*CD14, ITGAM, TMEM119, CD33*) in the glioblastoma dataset. Dashed line indicates the HSPC cluster defined in (**c**). **e** SingleR annotation of HSPC subsets in glioblastoma (GBM), tumor-free brain (TFB), bone marrow (BM), and blood samples magnetically enriched by CD34+CD45+ are shown as UMAP projections. Stacked barplot indicates fractions of HSPC subsets as annotated in the four datasets with absolute numbers shown within bars. *p* value determined by a two-tailed Fisher's exact test. HSC Hematopoietic stem cell, MPP Multipotent progenitor, CMP Common myeloid progenitor, GMP Granulocyte–monocyte progenitor, CLP Common lymphoid progenitor, MEP Megakaryocyte–erythroid progenitor. **f** The left panel shows UMAP projection of the integrated dataset used for cell cycle and differential gene expression analysis. Cells derived from each sample are displayed by different color-coding. Right panel: singleR annotation of HSPC subsets in the integrated dataset are highlighted on the UMAP plot by the the corresponding colors shown in the legend. NK cells Natural killer cells. **g** UMAP plot of cycling and non-cycling cells computed by Seurat. Stacked barplots show the proportion of cycling and non-cycling HSPC subsets in the glioblastoma (GBM), bone marrow (BM), and blood sample. *p* value determined by a two-tailed Fisher's exact test corrected by the Benjamini Hochberg procedure. **h** Heatmaps show a selection of top differentially expressed genes between bone marrow-resident and circulating HSPC subsets vs tumor-associated HSPCs as computed by MAST. The complete dataset is provided in the Supplementary Data 5–8. Source data are provided as a Source Data file.

(AHNPs)[56,57] cultured in the presence of HSPC-conditioned or control media. In this model, AHNPs showed a preferential migration towards HSPC conditioned media (*p* < 0.05), indicating a potential effect of tumor-associated HSPCs on NSC migration and recruitment in glioblastoma (Supplementary Fig. 6)

Patient-derived organoids are increasingly recognized as robust preclinical models for the study of cancer and response to therapy[58,59]. We cultured pure tumor cells from 3 primary glioblastoma patients and grew 3D organoids in the presence or absence of HSPCs, using a previously established protocol[60]. In two cases, 3D organoids could be maintained for >3 weeks in culture. As early as on day 4, we observed a significant increase in colony-forming activity of glioma cells co-cultured with HSPCs compared to controls (Fig. 5f, g). Furthermore, on day 10 after seeding, colonies in the presence of HSPCs formed long interconnections reminiscent of microtube networks reported by Oswald et al.[61] (Fig. 5f). Organoids supplied with HSPCs grew significantly larger when compared to cultures without HSPCs (Fig. 5h). Interestingly, patient-derived glioblastoma cells could stably maintain HSPC subsets for at least 20 days, in contrast to HSPC cultures seeded in the absence of tumor cells (Fig. 5i). These data suggest favorable conditions for the maintenance of HSPCs in glioblastoma. In addition, we noticed an expansion of a CD45+CD34− immune cell population within the organoid cultures, indicating that a subset of HSPCs are differentiating in the presence of patient-derived glioblastoma cells (Fig. 5i). However, PD-L1 expression was not differentially regulated in these experiments (Supplementary Fig. 7). To further characterize the relationship of HSPCs and glioblastoma cells during organoid expansion, we used a multiplex enzyme-linked immunosorbent assay (ELISA) to investigate the conditioned media for 30 different cytokines and growth factors. Interestingly, after 20 days of co-culture, we detected a significant increase of tumor-promoting cytokines such as interleukin 6 (IL-6)[62] and IL-8[63] (*p* = 7.5 × 10⁻⁷ and *p* = 0.0028, respectively), or positive regulators of immunosuppression such as chemokine ligand 2[64] (CCL2, *p* = 0.035) when compared to cultures containing tumor cells or HSPCs alone (Fig. 5j). In addition to these cytokines, we also detected a significant increase of soluble tumor necrosis factor α receptor 1 (sTNF-R1) (*p* = 0.004) and CCL4 (*p* = 0.006), after 9 and 20 days respectively. Other cytokines that were detected in the supernatants by this assay did not display significant changes between the different culture conditions (Supplementary Fig. 8).

In summary, using three in vitro and two ex vivo models, we observed consistent increase in tumor cell proliferation when cells were co-cultured with HSPCs. In T98G and LN229 cells, we detected a concurrent increase of PD-L1 expression on a

subpopulation of proliferating cells. HSPC-conditioned media promoted migration of AHNPs in vitro and co-cultures with patient-derived glioblastoma cells induced the secretion of tumor-promoting cytokines such as IL-6 and IL-8 or the immunosuppressive-related chemokine CCL2, indicating a potential role of HSPCs in promoting both, immunosuppression and malignancy phenotypes during glioblastoma progression.

**Tissue-associated HSPCs predict patient's survival, correlate with hematopoietic niche factors and immunosuppressive markers.** Next, we applied Syllogist to test the association of cellular composition with clinical outcome of 159 glioblastoma patients with follow-up clinical data available (TCGA)[31]. All tumor tissues were derived from previously untreated, primary glioblastoma patients undergoing standard surgery, radio- and adjuvant temozolomide therapy. Variables known to be associated with survival and therapy responses such as $O^6$-Methyl-guanine-DNA Methyltransferase (*MGMT*) promoter methylation[65], *IDH* mutation without 1p/19q codeletion[66] and biological subtypes[67,68] were also included in the analysis. By applying the random forest classifier[69], which can be employed to select for the best predictive variables within our dataset, we surprisingly identified three HSPC subsets, namely HSCs, CMPs, and promyelocytes among the most important predictors for overall survival of glioblastoma patients (Fig. 6a). The variable importance of these HSPC subsets was comparable with the above-mentioned positive controls.

Kaplan–Meier estimator of HSC^high and HSC^low patients and univariate Cox regression confirmed that HSC signals were negatively associated with overall and progression-free survival (Fig. 6b–d). To adjust for potential confounders such as age, *MGMT* methylation and *IDH* mutations, we fitted a multivariable Cox regression model of HSC signals for both, overall and progression-free survival. This model showed again a significant association of HSC with overall survival. In particular, our result indicated that in the TCGA cohort, at a given instant in time, a glioblastoma patient exhibiting an HSC signal ≥0.54 was 88% as likely to die as someone showing an HSC signal <0.54, adjusting for age, *MGMT* promoter methylation and *IDH* mutations (Fig. 6e and Supplementary Fig. 9). In our analysis, Syllogist did also detect a weak association of macrophages with overall survival, but, this result was not significant after correction for multiple testing (Supplementary Data 9).

In addition to survival, HSPC subsets significantly associated with signals from differentiated myeloid and lymphoid cell types (Supplementary Fig. 10). Interestingly, HSC^high glioblastoma samples significantly associated as well with increased expression levels of *TGFB1* and *IL10*, two genes coding for classical immunosuppressive

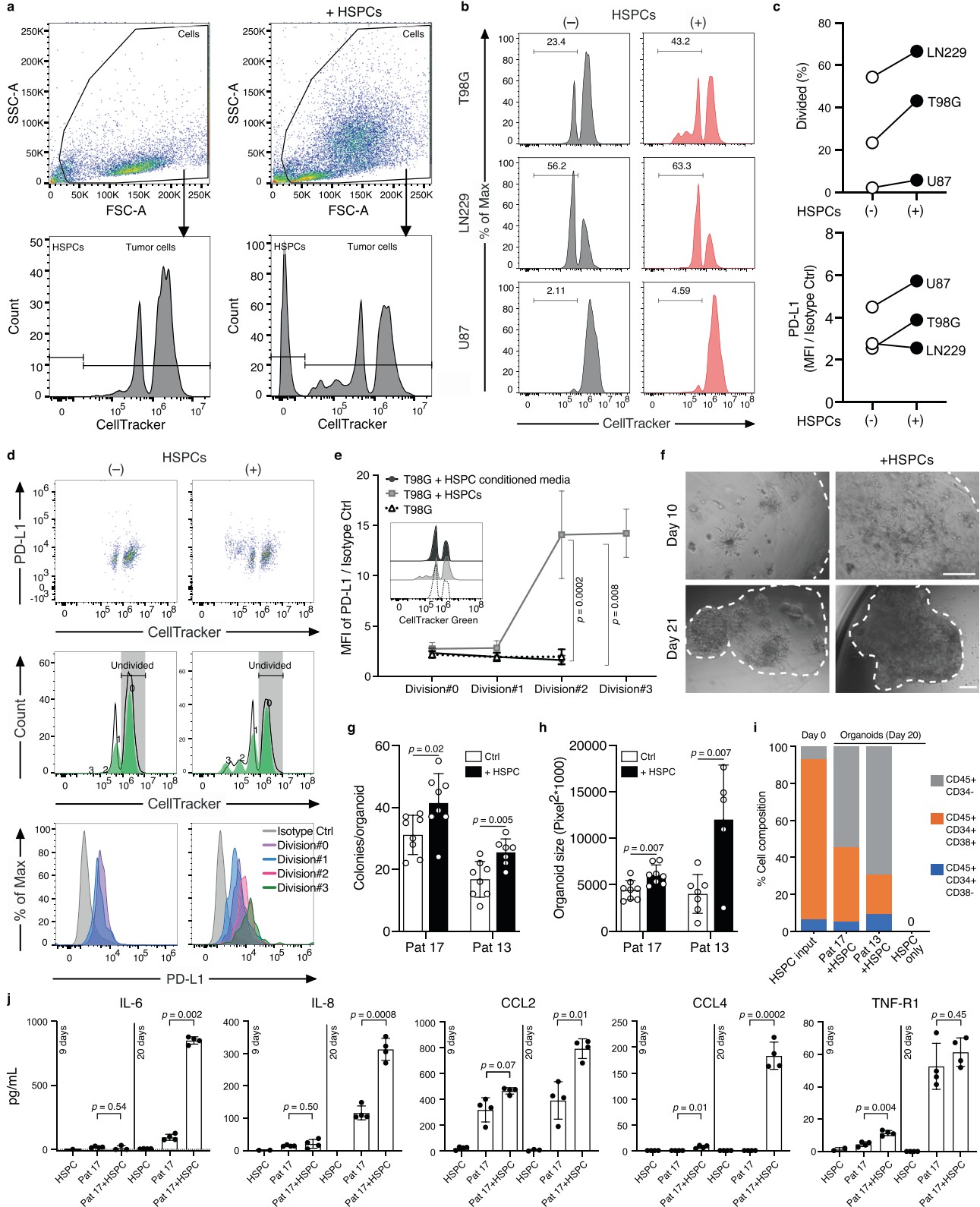

cytokines ($p = 1.4 \times 10^{-7}$, and $p = 9.7 \times 10^{-12}$ respectively, Student's t-test), but not with expression of proinflammatory cytokine genes such as IL2, IFNG, IL12 or IL17. An exception to this were TNF and IL6[62,70], which positively associated with the HSC[high] samples (Fig. 6f and Supplementary Fig. 11a). Furthermore, HSC[high] glioblastoma samples associated with the expression of immune checkpoint molecules, including PD-1 (PDCD1, $p = 2.7 \times 10^{-4}$), PD-L1 (CD274,

$p = 0.016$), and PD-L2 (PDCD1LG2, $p = 0.003$) (Fig. 6g). Moreover, HSC[high] glioblastoma samples also exhibited significantly higher expression of chemokines such as IL8, CCL2 and CCL4, in agreement with our organoid co-culture experiments shown in Fig. 5j (Supplementary Fig. 11b) or niche factors such as CXCL12, LEPR (leptin receptor), and FN1 (fibronectin) (Fig. 6h)[32]. These results demonstrated that, in a dataset of 159 patients, tumor-associated

**Fig. 5 HSPCs promote glioblastoma cell proliferation and PD-L1 expression. a** Representative flow cytometry profiles of T98G cells in presence or absence of Hematopoietic stem and progenitor cells (HSPCs) with CellTracker Green staining and gating strategy used to distinguish glioblastoma cells from co-incubated HSPCs. **b** Representative histogram of tumor cell proliferation of T98G, LN229, and U87 glioblastoma cell lines, co-cultured with/ without bone marrow-derived HSPCs (ratio = 1:1) for 48 h. Tumor cell proliferation was assessed by CellTracker Green CMFDA dilution measured by flow cytometry. **c** Comparison of the percentage of cells that underwent at least one cell division (% Divided), and comparison of the tumor surface PD-L1 expression determined by the fold-changes of Median fluorescence intensity (MFI) from Isotype Control (Ctrl). (n = 3 cell lines, one representative experiment of two shown. (**d**) Representative flow cytometry profiles (out of three experiments) using combined stain of CellTracker Green CMFDA and PE-PD-L1 in T98G cells co-cultured with/without HSPCs. **e** PD-L1 expression in T98G cells co-cultured with HSPCs, HSPC conditioned media or control media for the indicated cell divisions. Results are presented as mean ± standard deviation (n = 3–6 technical replicates, one representative experiment of two shown). Two-tailed unpaired Student's t-test. Inset: representative flow cytometric profile of CellTracker Green CMFDA staining for the three conditions tested. **f** Representative images of organoids from patient 17 (Pat 17) cultured with and without HSPCs at day 10 and 21 post seeding, scale bar = 100 μm. **g** Barplot represents number of colonies/organoid measured on day 4 (Pat 17) and day 14 (Pat 13) post seeding. **h** Barplot represents organoid size measured on day 14 (Pat 17) and day 21 (Pat 13) after seeding. In (**g**) and (**h**), p values determined by two-tailed, unpaired Student's t-test. Results are presented as mean ± standard deviation (n = 5–8 organoids for each condition and time point). Three patients tested, one patient excluded from this analysis as we did not achieve sustained growth. **i** Maintenance of HSPC phenotype in organoid culture alone or in co-culture with patient tumor cells (Pat 17 and 13), a representative experiment of two is shown. **j** Barplots represent cytokine concentration (pg/mL) measured in conditioned media of organoids derived from patient 17 in the presence or absence of HSPCs, or in HSPCs cultured alone in 3D Matrigel. Conditioned media were collected after 9 and 20 days. Data are presented as mean ± standard deviation, n = 1–4 technical replicates from one representative experiment of two. p values determined using unpaired, two-tailed Student's t-test corrected with the Benjamini–Hochberg procedure. IL-6, IL-8 Interleukin-6 and -8, CCL2, CCL4 CC-chemokine ligand 2 and 4, TNF-R1 tumor necrosis factor receptor 1. Source data of (**c**), (**e**), (**g**), (**h**), (**i**), and (**j**) are provided as a Source Data file.

HSPC subsets are predictive for clinical outcomes in glioblastoma, associate with an immunosuppressive phenotype, with hematopoietic niche factors and with specific cancer-promoting cytokines and chemokines.

## Discussion

Using Syllogist, we could determine the relative abundance of 43 different cell types in glioblastoma tissues using a gene enrichment approach. While other methods may have superior precision in the quantification of cell type proportions within a sample[71] they are restricted to a limited number of cell types. Syllogist can be useful in detecting multiple cell types, including hematopoietic progenitors, leveraging on a robust and validated reference transcriptome dataset[15].

Intending to characterize the cellular landscape of brain tumor tissues using a systematic and unbiased computational method, we identified HSPC transcriptomic signatures as markedly associated with brain tumors compared to normal brains and significantly enriched in glioblastoma when compared to lower grade *IDH* wildtype astrocytomas. Through a series of bioinformatic, flow cytometric, immunohistochemical and functional assays, we validated our initial working hypothesis and determined that HSPCs are infiltrating brain tumor tissues for the large proportion as immature progenitors. This is at first glance surprising because the bone marrow is the primary site of hematopoiesis in the adult. However, extramedullary hematopoiesis, especially in the liver and spleen, can sometimes be detected as a conserved physiological mechanism to maintain immunity under chronic anemias and myeloproliferative disorders[72]. Extramedullary HSPCs have also been detected under physiological conditions in several murine organs[73,74]. In addition, intravenously-injected HSPCs efficiently migrate to and infiltrate experimental rat[75] and mouse[76] gliomas, possibly by a CXCL12-dependent mechanism. Collectively, these data, together with our observations, may explain the enrichment of HSPCs in glioblastoma

Our findings suggest that HSPC subsets in brain tumors are positively associated with immunosuppressive and tumor-promoting phenotypes and negatively associated with patient survival. It is known that during cancer progression, bone marrow-derived HSPCs commit preferentially towards immunosuppressive lineages such as MDSCs induced by the tumor-derived cytokines granulocyte-macrophage colony-stimulating

factor (GM-CSF) and granulocyte colony-stimulating factor (G-CSF)[77]. Tumor-associated HSPCs may, therefore, be instructed by malignant cells to differentiate towards immunosuppressive myeloid cells. However, in murine models testing the effect of intravenously injected HSPCs, these cells were shown to replace local immunosuppressive myeloid cells with antigen-presenting cells, resulting in cytotoxic anti-tumor responses and tumor eradication[78]. In Flores *et al.*[79], ectopic injection of Lin⁻CCR2⁺ myeloid progenitors exhibited specific tropism to brain tumors and differentiated into antigen-presenting cells, cross-presenting to T cells in secondary lymphoid organs. These data indicate that tumor-associated HSPCs display a remarkable impact on the immunoregulation of the glioblastoma microenvironment: Contrary to observations in animal models that report favourable outcomes of intravenously injected HSPCs, our data reveal rather a cancer-promoting phenotype of the endogenous HSPCs that populate the human glioblastoma microenvironment. Therefore, remodeling of the lineage fate of tumor-associated HSPCs in humans may represent a potential therapeutic strategy to overcome immunosuppression and to provide the essential microenvironment for targeted immunotherapies. For example, blockade of the colony-stimulating factor (CSF) 1 - CSFR1 axis interfered with the maturation of bone marrow-derived hematopoietic progenitors into immunosuppressive myeloid cells[80] and reduced the pool of immunosuppressive myeloid cells in the brain[81]. Further, combining CSFR1 inhibition with PD-1/PD-L1 blocking antibodies resulted in superior tumor control compared to checkpoint inhibition alone in a mouse model of spontaneous neuroblastoma[82].

Our data suggest the requirement of cell-to-cell contact between HSPCs and glioblastoma cells for increased tumor cell proliferation and PD-L1 expression. Furthermore, patient-derived organoids revealed that glioblastoma cells can maintain HSPCs for extended periods of time in vitro, resulting in the secretion of cytokines and chemokines previously shown to promote tumor progression. These findings support the in vivo observations on the association with patient outcomes and immunosuppressive phenotypes.

However, our study presents with some limitations: Detailed investigation is needed to further understand the molecular mechanisms of our observations and to determine the role of the subpopulations responsible for the monitored effects. It is conceivable that some of the observed phenotypes are mediated by

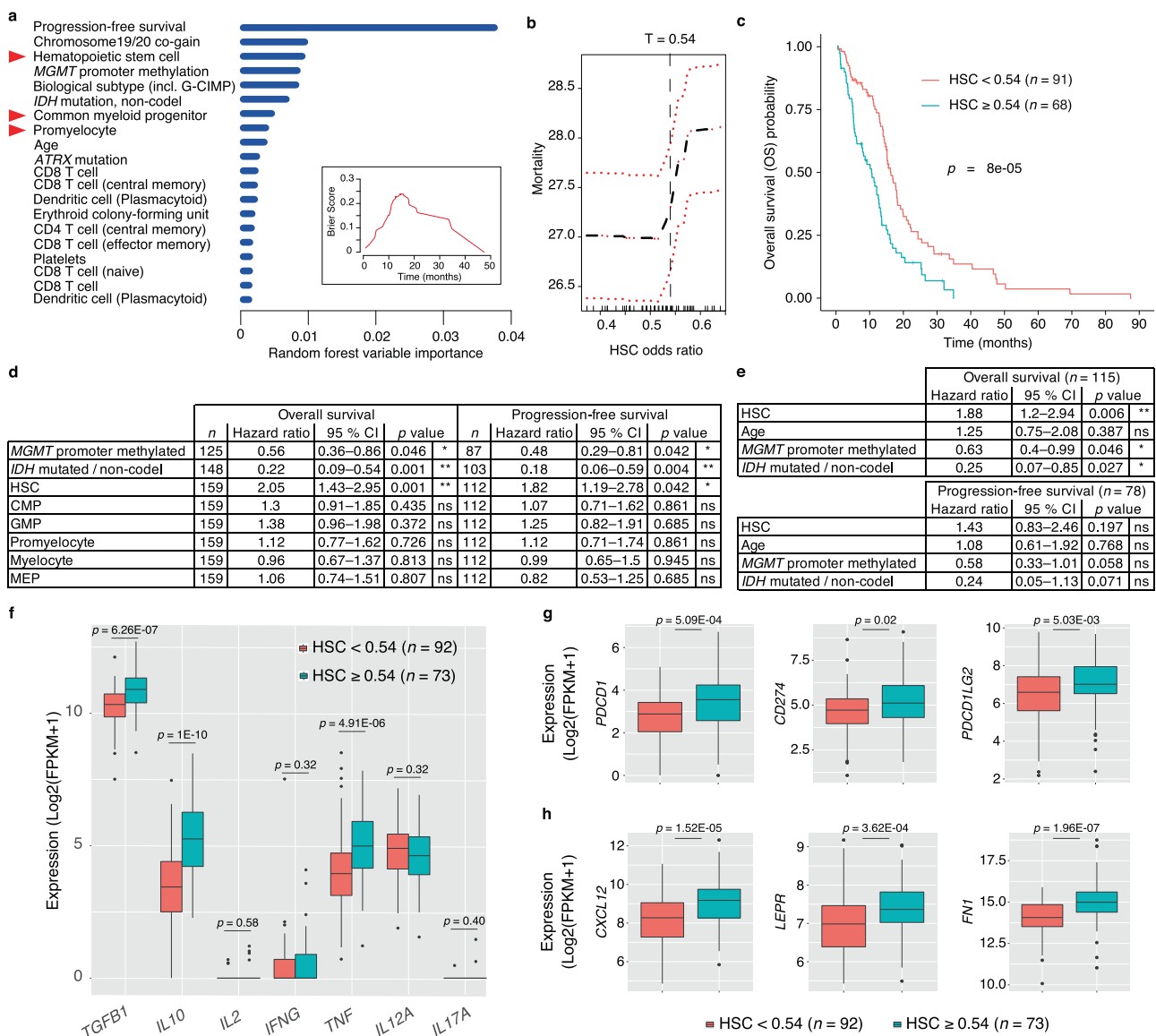

**Fig. 6 HSPCs predict clinical outcomes in glioblastoma and associate with an immunosuppressive phenotype and stem cell niche factors. a** Bar plot representing the top 20 predictors of glioblastoma overall survival in decreasing order of importance computed by the random forest classifier. Red triangles highlight HSPC subsets. Inset, Brier score indicates error rate of random forest classifier results as function of survival time (Brier score 0 = 0 % error, 1 = 100% error). MGMT O(6)-Methylguanine-DNA methyltransferase, G-CIMP Glioma CpG island methylator phenotype, IDH Isocitrate dehydrogenase. Source data are provided as a Source Data file. **b** Mortality rate as a function of hematopoietic stem cell (HSC) odds ratios derived by Syllogist and used to determine threshold separating HSC[high] (n = 63) and HSC[low] (n = 76) patients. Red lines represent 95% CI. **c** Kaplan–Meier plot of HSC[high] and HSC[low] patients using the threshold T = 0.54 selected from (**b**). Two-tailed logrank test. **d** Univariate Cox regression analysis of selected variables for overall survival and progression-free survival data. **e** Multivariable Cox proportional hazards model of the HSC subset and potential confounders (Age, *MGMT* methylation, and *IDH* mutations) for overall and progression-free survival. In (**d**) and (**e**), we used a two-tailed likelihood-ratio test corrected by the Benjamini–Hochberg procedure. All variables satisfied the proportionality hazards assumption (Methods). CMP Common myeloid progenitor, GMP Granulocyte–Monocyte progenitor, MEP Megakaryocyte–Erythroid progenitor. **f** Boxplots represent the expression of pro- and anti-inflammatory cytokines in HSC[high] (n = 73) and HSC[low] (n = 92) patient samples. TGFB1 Transforming growth factor beta 1, IL10, IL2 Interleukin 10 and 2, INFG Interferon gamma, TNF Tumor necrosis factor, IL12A Interleukin 12 subunit alpha, IL17A Interleukin 17A. **g** Boxplots represent the expression of the immune checkpoint markers PD-1 (*PDCD1*), PD-L1 (*CD274*), and PD-L2 (*PDCD1LG2*) in HSC[high] and HSC[low] samples. **h** Boxplots represent the expression of hematopoietic stem cell niche factors C-X-C motif chemokine 12 (*CXCL12*), leptin receptor (*LEPR*), and fibronectin (*FN1*) in HSC[high] and HSC[low] samples. In (**f–h**), boxplots are drawn with boxes representing the interquartile range (IQR), a line across the box indicating the median, and whiskers indicating 1.5 × IQR. Outliers are shown as closed dots. *p* values determined using a two-tailed Wilcoxon–Mann–Whitney U test corrected with the Benjamini–Hochberg procedure.

immune cells differentiating from local tumor-associated HSPCs and not by HSPCs themselves. In addition, our single cell data, while confirming presence of HSPCs and providing insights on their transcriptional profiles, represents a pilot study requiring follow up studies with a larger cohort.

In conclusion, the presence of multipotent HSPCs within the brain cancer microenvironment allows unconventional and straightforward access to an otherwise restricted immune compartment. Direct modulation of the lineage fate of tissue-associated HSPCs may represent a significant therapeutic

strategy to overcome immunosuppression or glioblastoma progression and warrants further studies. Flow cytometry-based analysis of hematopoietic progenitors in fresh tissue biopsies may furthermore serve as a prognostic factor in future clinical trials.

## Methods

**Cell type estimation using transcriptomes**. Cell type-specific signals were determined similarly to Cima I et al.[16]. First, we generated a map of specific genes for each cell type of interest using the primary cell atlas[15], a gene expression matrix containing information on $n = 20969$ gene transcripts. Technical replicates were averaged. For each gene, $g$, in each cell type, or 'lineage', $l$, a 'specificity index', $S$, was calculated based on Shannon entropy and the $Q$ statistics introduced by Schug et al.[18],

$$S_{(l|g)} = -\sum_{l=1}^{N} p_{(l|g)} \cdot \log_2(p_{(l|g)}) - \log_2(p_{(l|g)})$$

where $p_{(l,|,g)}$ is the relative expression of gene $g$ in lineage $l$. For each cell type, the top 80 genes with the highest specificity index ('specific genes') were selected and reported in Supplementary Data 1. This table represents the map of specific genes for each cell type of interest in decreasing order of specificity index. Next, for each RNA-Seq query sample, we predefined a threshold to define the set of expressed genes. Next, in the query RNA-Seq list of expressed genes, we counted the occurrences of the top 80 specific genes for each cell type present in our map of specific genes. To determine if the number of enriched genes was different from enrichment by chance, we generated 1,000 lists of 80 randomly selected genes from a comprehensive list of human genes derived from our reference transcriptomes and counted the average number of genes present by chance in each experimental RNA-Seq profile for each cell type. Finally, for each cell type in each experimental sample, a Fisher's exact test was applied to determine whether the number of enriched specific genes was equal to the number of randomly enriched genes. The resulting odds ratios were used in intersample comparisons to generate hypotheses on the differential content of cell types present in bulk tissues. In some experiments, we benchmarked Syllogist with previously published algorithms using TIMER 2.0 with default parameters[83].

**Tissue collection**. Biosamples were obtained from 29 patients after informed consent at the Departments of Neurosurgery of the University Hospitals Bonn and Essen. At each site, the local ethics committees approved the study (University Bonn #182/08; University of Duisburg-Essen, #19_8706_BO). Human biological samples and related data collected in Essen were provided by the Westdeutsche Biobank Essen (WBE, University Hospital Essen, University of Duisburg-Essen, Germany, approval 19_WBE_074). Baseline data for all patients are listed in Supplementary Data 4.

**Immunohistochemistry/immunofluorescence studies**. CXCL12 immunohistochemistry was performed on formalin-fixed, paraffin-embedded glioblastoma tissues obtained at the time of surgery. For antigen retrieval, slides with 2 µm-thick sections were pretreated boiling in sodium citrate buffer (pH = 6.0) for 30 min at 100 °C. Anti-CXCL12 antibody (Abcam ab9797, 1:600) was used to detect CXCL12 protein expression, and antibody-bound CXCL12 was then detected using the chromogen 3,3′-diaminobenzidine (DAB). Staining intensity was scored using a four-point scale from 0–3: 0 = no staining; 1 = cells weakly positive; 2 = cells moderately positive; 3 = cells strongly positive.

CD34 / CD45 immunofluorescence analysis was performed on formalin-fixed, paraffin-embedded glioblastoma tissues obtained at the time of surgery. 2 µm FFPE tissue sections were pretreated as described above. Anti-CD34 (Leica Biosystems, NCL-L-END, 1:250) and anti-CD45 (Abcam, ab10559, 1:250) antibodies were incubated for 1 h at room temperature and slides washed three times before the respective secondary antibodies (cross-adsorbed anti-rabbit Alexa 488 and anti-mouse Alexa 555, 1:800, Life Technologies) were applied for 1 h at room temperature. Slides were mounted with Vectashield Antifade Mounting Medium with DAPI (Vector Laboratories) before imaging on a ZEISS ApoTome.2 Microscope (Zeiss) with the Zeiss ZEN 2.3 Imaging Software.

**Tissue dissociation**. Fresh surgical tissue was placed in ice-cold Dulbecco's Modified Eagle Medium (DMEM)/F12-based transport medium in the operating room and received on ice at the lab within 30 min thereafter. The tumor tissues were subsequently cut into small pieces and homogenized in Iscove's Modified Dulbecco's Medium (IMDM) with 0.11 DMC U/mL neutral protease (Nordmark Biochemicals) at 37 °C for 1–2 hour in a shaker-incubator. The homogenized tissues were centrifuged for 10 min at 300 g, resuspended in IMDM and filtered through a 40 µm cell strainer for the following FACS and CFC assays.

**Flow cytometry**. Tumor cell suspensions were incubated for 5 min with human Fc-gamma receptor (FcR)-binding inhibitor (1:50; BioLegend) and assayed for Hematopoietic Stem Cells (HSC: 7AAD−Lin−CD34+CD38−CD45RA−CD90+), Multi-Potent Progenitors (MPP: 7AAD−Lin−CD34+CD38−CD45RA−CD90−), Multi-

Lymphoid Progenitors (MLP: 7AAD−Lin−CD34+CD38+CD45RA+CD90−), Common Myeloid Progenitors and Megakaryocyte–Erythroid Progenitors (CMP-MEP: 7AAD−Lin−CD34+CD38+CD45RA−CD10−), Granulocyte–Monocyte Progenitors (GMP: 7AAD−Lin−CD34+CD38+CD45RA+CD10−), and B-NK Progenitors (B-NK: 7AAD−Lin−CD34+CD38+CD45RA+CD10+). For exclusion experiments of potential non-hematopoietic contaminants, tumor cell suspensions were additionally assayed for CD45 (PE/Cy7, 1:50, BioLegend). The immunofluorescent monoclonal antibodies BV421-CD10 (1:50), BV510-CD90 (1:50), BV711-CD135 (1:50), BV785-CD45RA (1:50), PE-CD34 (1:25), FITC-CD144 (1:25) and anti-Human Lineage Cocktail 1 (Lin 1, 1:25) were purchased from BD Biosciences. The APC-CD38 antibody (1:50) and 7-AAD (1:20) were purchased from eBioscience. For analysis of in vitro experiments, tumor cell lines or cells from organoid experiments were assayed for PE-PD-L1 (1:100, BioLegend), BV510-CD45 (1:20, BioLegend) and BV786-CD56 (1:20, BD Biosciences) respectively, after co-culture with HSPCs. All the samples were analyzed on a FACS Celesta flow cytometer (BD Biosciences) using the FACS Diva v 8.0.1.1 software (BD Biosciences) and flow cytometry data were analyzed using FlowJo software, version 10.6.0 (Tree Star).

**Colony-forming cell (CFC) assay**. To observe hematopoietic colony-forming unit (CFU) formation, the cell suspension obtained from tumor tissue was seeded in methylcellulose media: (MethoCult H4230 and MethoCult SF H4236, Stemcell Technologies) according to manufacturer's protocol. Both media were supplemented with IL-3 (20 ng/mL), IL-6 (20 ng/mL), G-CSF (20 ng/mL), GM-CSF (20 ng/mL), SCF (50 ng/mL) and erythropoietin (3 units/mL). After incubation for 14–16 days at 37 °C with 5 % $CO_2$, the colonies were characterized and scored according to their morphology on a ZEISS AX10 Inverted Microscope (Zeiss).

**Single Cell RNA Sequencing and analysis**. CD34+ and CD45+ cells from two fresh glioblastoma tissues, one tumor-free region tissue, healthy bone marrow monoclonal cells (CD34+ Lonza, 2M-101A) and one healthy PBMC sample (Lonza, 4W-270) were used for scRNA-Seq studies. Tissue samples were dissociated as previously described. CD34+/CD45+ positive magnetic selection was performed using the REAlease® CD45 (TIL) MicroBead Kit (Miltenyi Biotec, 130-121-563) and, immediately after removal of the CD45 complex, using the CD34 MicroBead Kit UltraPure (Miltenyi Biotec, 130-100-453) on the CD45 positively-selected samples, according to the manufacturer's instructions. After isolation, samples were stored at −80 °C in freezing medium (15% DMSO and 20% FBS in IMDM) until further processing. Before library preparation, samples were inspected for dead cells using trypan blue exclusion. At this stage, one glioblastoma sample was excluded from further analysis because of the presence of multicellular aggregates and 20% trypan blue positive cells. All other samples (Pat 24, Pat 25, bone marrow and PBMC sample) contained >92% viable cells without doublets, and were used for single cell sequencing. Next, library preparation was performed with all samples using the Chromium Next GEM Single Cell 3' Reagent Kits v3.1 (10x Genomics). Appropriate volume for the recovery of 770 cells was loaded onto a chip and DNA libraries were prepared according to the manufacturers protocol. Quality control of prepared libraries was performed using the Agilent 2100 Bioanalyzer prior to sequencing. Paired-end sequencing of all libraries was performed using the Illumina NovaSeq 6000 system on one flow cell lane. Illumina basecall (.bcl) data were converted and demultiplexed to FASTQ files using the bcl2fastq v2.20 software. Read alignment to the hg38 human reference genome, counts and cell-calling were computed using the 10x Genomics Cell Ranger 4.0.0 pipeline[84] for each sample with the "cellranger count" command and default parameters. Median UMI counts per cell in all samples ranged from 14,188 to 19,610. Count data were analyzed using Seurat[85] (v4.0). First, we removed cells with low counts (nFeature_RNA < 200) and high% of mitochondrial genes (>15%). Data were subsequently log-normalized before further analyses. Clustering was computed using the FindClusters function with parameter "resolution" set at 0.5. UMAP was computed using the first 30 dimensions as input. Annotation of cell types was performed using SingleR 1.4.1 with default parameters and the BlueprintEncodeData reference obtained from the celldex 1.0 package. The glioblastoma, bone marrow and blood samples were then integrated in one dataset for UMAP plotting, cell cycle and differential gene expression analysis. To this end, we used the default Seurat workflow on the log-normalized data and the IntegrateData function with the first 50 dimensions as input. Cell cycle analysis was performed using the CellCycleScoring function and the default list of cell cycle genes provided by Seurat 4.0 (cc.genes). Differential gene expression (DGE) was performed on the integrated dataset using the normalized and scaled data and the MAST[86] algorithm provided within the FindMarkers function with default parameters. For each HSPC subset, we selected genes with adjusted $p$ value <0.05 that were commonly regulated between the glioblastoma-bone marrow and the glioblastoma-blood sample DGE analyses. The complete DGE results are reported in Supplementary Data 5–8.

**In vitro HSPCs and tumor cell co-culture**. Tumor cell lines T98G, LN229, and U87 (ATCC), were labeled with CellTracker Green CMFDA (5-chloromethyl-fluorescein diacetate, Thermo Fisher Scientific) at a final concentration of 1 µM for 15 min at 37 °C in darkness. After two washes with DMEM supplemented with 10 % FBS, the labeled tumor cells were combined with enriched bone marrow-derived HSPCs (Lonza Bioscience) in cell culture plates at 1:1 ratio or with the conditioned

medium derived from HSPC culture. After 48 h of co-incubation, supernatants were gently removed from the cell culture suspension and adherent tumor cells were detached with 0.11 DMC U/mL neutral protease (Nordmark Biochemicals) at 37 °C for 10 min and collected for immunofluorescent staining with PE-PD-L1. Flow cytometry analysis was performed to distinguish tumor cells from HSPCs and to monitor PD-L1 expression and proliferation in tumor cells by CMFDA dilution.

**Patient-derived organoid co-culture.** For organoid co-culture, patient-derived glioblastoma tumor cells (P13, P16, P17) were treated with 0.11 DMC U/mL neutral protease (Nordmark Biochemicals) at 37°C for 10 min, centrifuged for 5 min at 400 g, and resuspended in Neurobasal-A medium (Life Technologies). After mixing with bone marrow-derived HSPCs at 1:1 ratio, cell suspensions were added into 4 times volume of Matrigel (Corning) in a separate tube kept on ice, and further transferred into a 96-well "droplet- forming plate" at a density of 2,000 cells per 20 μL, similarly as described in[60]. Each droplet was then transferred into an individual well of a 96-well plate and maintained in Neurobasal-A medium supplemented with 1% or 10% FBS, 50 U/mL penicillin, 50 mg/mL streptomycin, 0.5 mM glutamine, 10 μg/mL FGF, 10 μg/mL EGF at 37 °C with 5 % $CO_2$. Medium was exchanged every 2 days.

**Multiplex ELISA array.** Conditioned media from patient-derived organoids were assayed quantitatively for the following proteins: BDNF, CCL11, CCL17, CCL2, CCL24, CCL26, CCL4, CCL5, CNTF, CSF2, CXCL8, EGF, FAS, GDNF, IFNG, IL10, IL18, IL1A, IL1B, IL4, IL6, LIF, MMP2, MMP3, NGF, TGFB1, TIMP1, TNF, TNFRSF1A, VEGFA using a commercially available sandwich ELISA array kit (Quantibody® Human Neuro Discovery Array Kit, RayBiotech, QAH-NEU-1) according to the manufacturer's instructions and analyzed using the ImageJ Software v1.46r.

**Invasion assay.** To determine whether HSPCs recruit neural stem cells, a cell invasion assay was performed using Cytoselect™ 24 well collagen 1 colorimetric kit (Cell Biolabs). 250 μL cell suspension containing $0.5 \times 10^6$ cells/mL adult human neural progenitor cells (AHNPs[56,57]) were added to the upper chamber. Lower chambers were filled with 500 μL of CD34$^+$ HSPC (Lonza) control or conditioned media. The cells were incubated for 24 h, 48 h, and 72 h at standard cell culture conditions (37°C, 5% $CO_2$). Non-invasive cells were removed from the upper chamber and invaded cells were stained and quantified by colorimetric measurement as described in the manufacturer's protocol.

**Statistical analysis and random forest classifier.** Statistical analyses were performed in the R environment (version 3.6.1)[87] or using the Prism software (v 8.4, GraphPad). Paired (Fig. 1f) and unpaired samples were tested using two-tailed Student's t-test. p values were adjusted by the Benjamini–Hochberg procedure in the case of multiple comparisons with control of the false discovery rate (FDR) at the 5 % level. Associations between categorical data were assessed using two-tailed Fisher's exact test. Correlations were described using Pearson's r. Kaplan–Meier estimators were compared using the log-rank test. In Fig. 6d, association of Syllogist signals with survival data were assessed by comparing univariate Cox proportional hazards models using the likelihood ratio test (p values corrected by the Benjamini–Hochberg procedure). To this end, all continuous variables were binned into two categories each using appropriate thresholds. This was necessary for the proportional hazard assumption to be met for all variables included in the analysis. This assumption was tested for each variable by the Schoenfeld individual test before fitting the models. To adjust for potential confounders, we used Cox multiple regression models. Box plots were drawn with boxes representing the interquartile range (IQR), a line across the box indicating the median, and whiskers indicating $1.5 \times IQR$. The significance threshold was set at 0.05. For the random forest classifier we used the rfsrc function of the random forest package randomForestSRC with the following parameters: ntree = 1000, nsplit = 1, importance = "anti". Variable importance and estimation of the Brier score were reported in Fig. 6a.

**Reporting summary.** Further information on research design is available in the Nature Research Reporting Summary linked to this article.

## Data availability
References to repositories for publicly available RNA-Seq datasets analyzed during the current study are listed in Supplementary Data 10. Single cell RNA-Seq data generated in this study (Fig. 4 and Supplementary Fig. 5) are available at the Gene Expression Omnibus under the accession number GSE165238. The source data underlying Figs. 1–6 and Supplementary Figures 2 and 4–8 are provided as a Source Data file. All the other data supporting the findings of this study are available within the article and its supplementary information files and from the corresponding author upon reasonable request. Source data are provided with this paper.

## Code availability
R script and reference files used for transcriptome analyses are available at Zenodo with the identifier https://doi.org/10.5281/zenodo.4782282[88].

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

## Acknowledgements

The authors are grateful to all of the donors who participated in this study. We thank Mihaela Keller for processing of patient tissues and the High Throughput Sequencing Unit (Genomics & Proteomics Core Facility, DKFZ), for their sequencing services. Funding: The study was supported by a grant from the Wilhelm Sander Stiftung

(2017.148.1) to IC and BS, by a grant from the Deutsche Forschungsgemeinschaft DFG/GRC-CRU337 (SCHE656/2-1) to BS and the German Cancer Consortium (DKTK). J.T.S. is supported by the German Cancer Consortium (DKTK), the Deutsche Forschungsgemeinschaft (DFG) grant SI1549/3-1 (DFG/GRC-CRU337) and SI1549/4-1, and the German Cancer Aid (grant no. 70112505; PIPAC consortium).

## Author contributions

Bioinformatics and data processing: C.D., S.K., I.C. Immunofluorescence and immuno-histochemistry: I.L., C.D., S.T.H., T.B., I.C. Flow cytometry: I.L., C.D., V.U., I.C. Single cell RNA-Seq: C.D., A.S., V.U., J.C.B., I.C. Patient-derived ex vivo and in vitro assays: I.L., C.D., S.P., I.C. Clinical samples, clinical data and guidance on clinical aspects of the study: L.R., S.K., J.T.S., U.S., M.G., B.S. Manuscript draft: C.D., I.C. with input from I.L., L.R., S.T.H., V.U., S.P., S.K., J.T.S., B.S. Supervision, study design and funds acquisition: B.S., I.C.

## Funding

## Competing interests

J.T.S.: Bristol-Myers Squibb, Celgene, Roche (Research Funding - outside of this study); AstraZeneca, Bristol-Myers Squibb, Celgene, Immunocore, Novartis, Roche, Shire (Consulting or advisory role); AstraZeneca, Aurikamed, Baxalta, Bristol-Myers Squibb, Celgene, Falk Foundation, iomedico, Immunocore, Novartis, Roche, Shire (honoraria); minor equity in FAPI Holding and Pharma15 (<3%) and member of the Board of Directors for Pharma15 outside the submitted work. All other authors declare no competing interests.
