## [Peer Review File · Nature Communications]

REVIEWER COMMENTS

Reviewer #1 (Remarks to the Author); expert on bioinformatics and immunology:

General comments

In the manuscript by Lu and colleagues a computational method was developed to estimate different cell types from gene expression data and was applied on data from brain tissue samples. The authors identified hematopoietic and progenitor cells within tumor tissue and carried out additional experiments with cell lines and patient-derived organoids. Finally, using TCGA data the authors show that the tumor-associated HSPCs are prognostic for survival and correlate with immunosuppressive phenotypes.

Given the fact that glioblastomas are purely immunogenic, a comprehensive analysis and functional characterization of different infiltrating immune cells is of utmost importance in order to develop novel immunotherapies. In this light, the work is of potential interest to a broader audience.

However, there are several concerns regarding both, computational and experimental methodology that need to be addressed. First, there are a number of algorithms used to estimate different cell phenotypes from bulk RNA-sequencing data using either deconvolution approaches or gene enrichment analysis (see recent review PMID: 31515541). There is no evidence that the presented algorithm is superior since only in silico validation was carried out and there is now validation with experimentally generated data. Moreover, it appears that the authors used an outdated gene expression data (Reference 16) and it is thus questionable how useful is the newly developed method. Hence, thorough validation and benchmarking is necessary in order show that the presented method represent a novel contribution. Second, the authors did not make use of the available data from other publications that could confirm their observations. For example in the studies (PMID: 29628290, PMID: 29141660) TCGA data was analyzed with different algorithms and although the immune cell subsets are different, this analyses would support the observation of the authors. Of note, hypotheses can be generated also using inferior computational methods and if the biological finding is the major message of the study, than the manuscript has to be rewritten. Third, however, the validation of the observation has not been carried out using golden standard method, i.e. IHC or IF. The used marker CD34+ is not specific for hematopoietic cells but it is also expressed in other cell types (see PMID: 24497003). Thus, the presented results are neither robust nor there is strong evidence for the reported observation.

Specific comments

1. The authors should avoid using the term deconvolution as it is confusing: deconvolution is used for algorithms solving an inverse problem like in CIBERSORT whereas the presented method is basically gene enrichment analysis.
2. Page 20, first paragraph: The two references have the same number (19).

Reviewer #2 (Remarks to the Author); expert on glioblastoma:

This manuscript by Lu et al describes that bone marrow-derived hematopoietic stem and progenitor cells (HSC) are found in intracranial human glioblastoma (GBM). This is elegantly demonstrated via genomics and flow cytometry using both cell lines and patient-derived tumor cells. However, other studies have described that HSCs migrate to and are found in GBM. Comments are below:

1. The authors developed Sylogist, a new transcription deconvolution algorithm to identify specific signals from 43 different cell types. This was then applied to brain tumors and compared to normal

brain to acquire an immune gene signature in tumor tissues. Did the authors validate the frequencies of the cells identified using Syllogist in the brain tumor using other methods?

2. It would be helpful if the authors defined or stated the markers and genes used to make their cell type classifications perhaps even in the supplementary materials.
3. Figure 2 is very interesting, associating HSC abundance with tumor subtypes.
4. It is known that HSCs and neural stem cells respond to similar tropic factors. Did the authors consider that HSCs may be recruiting NSCs to contribute to tumor growth? And consider the role of HSCs in angiogenesis of tumors?
5. The CXCL12/CCR4 is a well know axis to recruit HSC, NSCs, T cells and cancer stem cells. Did the authors test other chemokine/ligand axes of potential migration to tumor? Please show this data.
6. The authors did an excellent job identifying stem and progenitor cell populations in tumor. However, they failed to address that HSPCs also differentiate into the immune cells found in the tumor including MDCs, TAMs, DCs. How do the HSPC numbers correlate with other myeloid derived progenitor populations known to differentiate from these cells and were other myeloid cell populations more strongly correlated with outcomes than HSPCs?
7. In the co-cultures where HSPC input was shown to enhance proliferation how stable in the Lin-HSPC phenotype during these cultures? Gliomas may drive differentiation of HSPCs to myeloid derived suppressor cells or other immature myeloid cells that may actually be contributing to tumor cell proliferation and not the HSPCs themselves. Do the investigators have evidence that HSPCs drive tumor growth or progeny derived from these cells?

These are important and interesting findings that highlight the recruitment of HSPCs in the natural progression of gliomas and potential for contribution the glioma development.

Reviewer #3 (Remarks to the Author); expert on HSPC:

In the present manuscript by Lu et al., the authors investigate the immunologic networks that are present in brain tumors and reveal that hematopoietic stem/progenitor cells (HSPCs) are present in the tumor microenvironment and correlate with the disease progression. More specifically, the authors establish a reference-based deconvolution algorithm (Syllogist) and determined a signature for 43 different cell types (with a special focus on immune cell types). Subsequently, they profiled the cellular landscape of brain tumor tissues and, intriguingly, found increasing signals from HSPCs. They also verified the presence of cells with an HSPC phenotype (morphological and functional) and showed that direct contact between HSPCs and tumor cells is required for enhancing their proliferation and PD-L1 expression. Finally, they make a correlation between HSPCs' numbers and the disease progression.

The authors have combined several different assays (bioinformatic, phenotypic, functional) to associate the presence of HSPCs with brain tumors progression. Overall, their findings are of great interest and importance with potential clinical applications. However, fundamental issues need to be addressed.

Major comments:

- In Figure 1, the authors should be more specific of how each population was defined (especially HSCs and markers used for isolation).
- In Figure 2: The authors present neural/glial as a common set of genes; however, they are known to have distinct signatures. The authors should add/split if one of them is more affected or in case they are both affected, they should comment on the clustering they did for the analysis. Moreover, based on Fig 2A, HSPCs seem to be expressed in healthy brains. The authors should comment more on whether HSC and/or HSPCs signature is found in the healthy brain (even if at much lower levels).
- In Figure 3D: More cells appear to be CD34+ and CD45high (apart from the asterisk, which is an endothelial cell). Which cell type is represented with this? In general, the authors should provide more evidence on the cellular composition of the tumors examined (including lower magnification images). Also, fundamental controls are missing: do healthy brains also contain HSCs? Less, more?
- In the same context, the authors state that tumor cells provide the necessary microenvironment for HSPC colonization (based on Fig. 3). One question rising is whether a higher number of HSPCs infiltrates the tissue or if they could proliferate more in the tumor microenvironment. Figure3F is

missing the quantification of CFUs derived from BM HSPCs. Are HSPCs from brain tumors equally potent? Also, further characterization from brain tumors HSPCs is required such as transcriptome compared to BM and brain healthy HSPCs, cell cycle analysis (Ki67 or Hoechst).

- Figure 4C: the authors should show FACS plots/gating strategy of HSPCs.
- They also correlate HSChigh glioblastoma with increased TGF- β 1 and IL-10 (related with immunosuppression) and not pro-inflammatory cytokines. They should provide more evidence regarding the immunosuppressive effect of HSChigh in the tumor environment. The authors should address this by using the in vitro assay of co-cultures and measure the profile of cytokines (ELISA).
- In Figure 4E: They perform co-cultures of HSPCs with patient-derived organoids and show that they display increased organoid size. The authors should also perform a cytokine array.
- In Figure 5: HSCs are usually a rare cell population. How do the authors explain such enrichment using brain tumors?
- Discussion: It has been previously shown in animal models that HSPCs can infiltrate the brain tumor environment, further supporting the authors' findings. These studies could also be implemented in the discussion (<https://www.ncbi.nlm.nih.gov/pmc/articles/PMC5101999/>).

Minor comments:

- Material and Methods (line 609): HSC and MPPs: CD34+CD38+ (it should be corrected to CD38-)
- Typo in line 500 (respons es)

Lu et al. Tumor-associated hematopoietic stem and progenitor cells positively linked to glioblastoma progression

POINT BY POINT RESPONSE TO REVIEWERS

We would like to thank the Reviewers for their constructive, insightful comments and suggestions on our work. We hope to have addressed the comments and questions of each of the Reviewers in the point by point response below and in the revised manuscript. All changes to the initial submission are reported here and highlighted in the accompanying manuscript in yellow color. References cited in this response to Reviewers are listed at the end of this document.

Reviewer #1

General comments

In the manuscript by Lu and colleagues a computational method was developed to estimate different cell types from gene expression data and was applied on data from brain tissue samples. The authors identified hematopoietic and progenitor cells within tumor tissue and carried out additional experiments with cell lines and patient-derived organoids. Finally, using TCGA data the authors show that the tumor-associated HSPCs are prognostic for survival and correlate with immunosuppressive phenotypes. Given the fact that glioblastomas are purely immunogenic, a comprehensive analysis and functional characterization of different infiltrating immune cells is of utmost importance in order to develop novel immunotherapies. In this light, the work is of potential interest to a broader audience. However, there are several concerns regarding both, computational and experimental methodology that need to be addressed.

Response: We thank this Reviewer for the comments and insights, especially on the computational method and validation steps reported in our paper.

[1] First, there are a number of algorithms used to estimate different cell phenotypes from bulk RNA-sequencing data using either deconvolution approaches or gene enrichment analysis (see recent review PMID: 31515541). There is no evidence that the presented algorithm is superior since only in silico validation was carried out and there is now validation with experimentally generated data.

Response: To address the Reviewer's concerns about the use of an experimentally generated dataset, we included in the revised manuscript 4 different datasets with experimental data available (2 flow cytometry datasets with paired RNA-Seq data used in the validation of xCell¹, and 2 immunofluorescence datasets with paired RNA-Seq data used for the validation of QuanTIseq²). We then used this data to test our method and benchmark with 4 other previously published deconvolution methods (CIBERSORT³, xCell, QuanTIseq and EPIC⁴). These data are now available in **Fig.1c, d** of the revised manuscript. The data confirm, for the cell types that are commonly detected among the 5 computational methods, a substantial similarity of our method with the benchmarked methods using intersample comparisons. Syllogist may represent, in our opinion, a complementary tool for cell type identification from RNA-Seq data rather than a superior method. Other methods may have superior precision in the quantification of cell type proportions within a sample, but they are restricted to a limited number of cell types. Syllogist, on the other hand, can be used to detect a larger number of cell types, including hematopoietic progenitors. In contrast to xCell, which uses a similar approach to ours, we used Shannon entropy⁵ for the identification of cell-type specific markers. We also focused on a single dataset to obtain reference signatures (the Primary Cell Atlas⁶). This has the advantage of increasing the number of gene transcripts that can be used to search for cell type specific signatures, compared to merging several datasets, which may lead to a loss of information due to casewise deletion of genes that are not commonly expressed between different datasets (e.g. our specific gene signatures were derived from a pool of $n = 20,969$ genes, while xCell gene signatures could be derived from $n = 10,808$ genes). Ultimately, we believe that Syllogist may be useful for the generation of working hypotheses that, however, need to be validated using experimental data. We have added a short paragraph in the **Discussion** of the revised manuscript to clarify this aspect and referenced the recent review that is kindly suggested by the Reviewer.

[2] Moreover, it appears that the authors used an outdated gene expression data (Reference 16) and it is thus questionable how useful is the newly developed method. Hence, thorough validation and benchmarking is necessary in order show that the presented method represent a novel contribution.

Response: For the reference expression matrix we used the Primary Cell Atlas⁶, published in 2013, which we agree is not novel. While newer datasets may provide more precise insights or expand the number of cell types analyzed, we decided to focus on a well-established and validated dataset that has been previously shown to generate robust cell type specific signatures. The Primary Cell Atlas has also been used as one of the reference datasets for cell type deconvolution in xCell¹. Our lab has previously used the same dataset as a discovery tool to identify circulating endothelial cells in colon cancer patients⁷ and a fibroblast subtype in lung cancer⁸. Therefore, we believe that the Primary Cell Atlas is a robust dataset that allows us to generate discovery-driven hypotheses. We recognize however the need for an updated reference transcriptome comprising ideally a large amount of cell types which may be available in the near future, e.g. through the human cell atlas initiative⁹.

[3] Second, the authors did not make use of the available data from other publications that could confirm their observations. For example in the studies (PMID: 29628290, PMID: 29141660) TCGA data was analyzed with different algorithms and although the immune cell subsets are different, this analyses would support the observation of the authors.

Response: We thank the Reviewer for this suggestion. Indeed, it would be interesting to compare our method with previously validated methods on the TCGA dataset. For this purpose, we analyzed the agreement of 5 computational methods, including the ones kindly suggested by the Reviewer in the analysis of common cell types in brain tumor tissue transcriptomes. For the cell subsets analyzed in bulk RNA-Seq from the TCGA GBM tissues, all methods broadly agreed within each other. In particular, our method agreed with EPIC (for 6/6 cell subsets) and xCell (5/6 cell subsets). For each cell subset analyzed, Syllolist was always in agreement with at least two other computational methods. These data are included in the revised manuscript as **Fig. 1e**.

[4] Of note, hypotheses can be generated also using inferior computational methods and if the biological finding is the major message of the study, than the manuscript has to be rewritten.

Response: The biological findings indeed represent the major message of the study, especially in the revised manuscript (e.g. we added a new result section with single cell characterization of tumor-associated HSPCs). However, we would not have come across the initial working hypothesis without the use of our computational method. As suggested by this Reviewer, we have adapted the manuscript accordingly and hope that the revised manuscript better reflects the overall message of the study.

[5] Third, however, the validation of the observation has not been carried out using golden standard method, i.e. IHC or IF. The used marker CD34+ is not specific for hematopoietic cells but it is also expressed in other cell types (see PMID: 24497003). Thus, the presented results are neither robust nor there is strong evidence for the reported observation.

Response: We agree with the Reviewer that CD34 marker alone is not sufficient for the detection of for hematopoietic progenitors. In the initial submission we have provided an immunofluorescence image of tumor-associated HSPC by the combination of markers CD34 and CD45. In the revised manuscript we have expanded on the immunofluorescence studies by providing a more thoroughly validated set of data. We performed the double-immunofluorescence in the Institute of Neuropathology using carefully validated protocols that are in part used for the diagnostic routine. This data is now presented in **Fig. 3a**. By using IF, CD34+CD45+ double positive cells were documented in 4/4 GBM samples analyzed.

In addition, we reported flow cytometry data and CFC assays commonly used in the analysis of HSPCs from bone marrow or blood (**Fig. 3c-f**).

Finally, in the revised version of the paper, we provide as well single cell RNA-Seq data of tumor-associated HSPCs, providing another experimental support for our initial working hypothesis (**Fig. 4 and Supplementary Fig. 5**). In conclusion, we have strived to test the initial hypothesis using multiple independent experimental techniques.

Specific comments

[6] *The authors should avoid using the term deconvolution as it is confusing: deconvolution is used for algorithms solving an inverse problem like in CIBERSORT whereas the presented method is basically gene enrichment analysis.*

Response: We have modified the manuscript according to the Reviewer's suggestion.

[7] Page 20, first paragraph: The two references have the same number (19).

Response: We have corrected the references accordingly.

Reviewer #2

This manuscript by Lu et al describes that bone marrow-derived hematopoietic stem and progenitor cells (HSC) are found in intracranial human glioblastoma (GBM). This is elegantly demonstrated via genomics and flow cytometry using both cell lines and patient-derived tumor cells. However, other studies have described that HSCs migrate to and are found in GBM. Comments are below:

Response: We appreciate the comments of the reviewer and are especially thankful for the insights and ideas for experiments related to HSPCs and their microenvironment. We have strived to address all questions and concerns of this Reviewer in the point by point responses below and have updated the manuscript accordingly. In the **Discussion**, we have cited the most important studies reporting on the migration of HSPCs in glioma models.

[1] The authors developed Syllogist, a new transcription deconvolution algorithm to identify specific signals from 43 different cell types. This was then applied to brain tumors and compared to normal brain to acquire an immune gene signature in tumor tissues. Did the authors validate the frequencies of the cells identified using Syllogist in the brain tumor using other methods?

Response: To perform this control suggested by the Reviewer, we tested the agreement between our method and 4 previously established computational methods (CIBERSORT³, xCell¹, QuanTIseq² and EPIC⁴) on the TCGA brain tumor dataset, by computing a Pearson correlation matrix on the resulting values obtained by these and our method. Our method agreed with at least two other computational methods for all the cell types commonly analyzed. In particular, our method agreed with EPIC (for 6/6 cell subsets) and xCell (5/6 cell subsets). These data are now included in the revised manuscript in **Fig. 1e**.

[2] It would be helpful if the authors defined or stated the markers and genes used to make their cell type classifications perhaps even in the supplementary materials.

Response: The information on the gene signatures used as reference for Syllogist can be found in **Supplementary table 1** of the revised manuscript.

[3] Figure 2 is very interesting, associating HSC abundance with tumor subtypes.

Response: We thank the Reviewer for this positive comment

[4a] It is known that HSCs and neural stem cells respond to similar tropic factors. Did the authors consider that HSCs may be recruiting NSCs to contribute to tumor growth?

Response: This is an interesting suggestion that we did not consider during the initial submission. We performed an invasion assay using adult human neural progenitors (AHNPs) derived from the hippocampus as a in vitro model for NSC^{10, 11}. In this experiment, we used HSPC conditioned or control media as potential chemoattractants. We observed, unexpectedly, that the HSPC conditioned media significantly and consistently promoted migration of NSCs through basement membrane coated chambers, in comparison to control media ($n = 3$ experiments). These data are now included in **Supplementary Fig. 6** and reported in the **Results** section of our manuscript.

[4b] And consider the role of HSCs in angiogenesis of tumors?

Response: We did not consider the role of HSCs in the angiogenesis of tumors and we thank the Reviewer for this suggestion. When performing immunofluorescence studies, we indeed noted that HSPCs were mostly located in close proximity to endothelial cells (e.g. **Fig. 3a** of the revised manuscript). In addition, when performing coculture experiments with patient-derived organoids in the presence of HSPCs, we observed a significant increase in chemokines that have been shown to promote angiogenesis in brain tumors, such as IL-8¹² and CCL2¹³ (**Fig. 5j**). We may therefore hypothesize that tumor-associated HSPCs can promote angiogenesis by inducing the release of proangiogenic factors in the tumor microenvironment. However, when comparing tumors with high or low content of HSPCs, we could not find significant changes in the expression of classical angiogenic or endothelial genes (see **Fig.1 for Reviewers only** below)

Figure 1 for Reviewers only: HSC signals in GBM patients do not associate with markers of angiogenesis or endothelial cells. Boxplots represent the expression of known genes associated with angiogenesis or representing tumor endothelial cells in HSC^{high} and HSC^{low} patients reported in Fig. 6. *p* values determined using Student's *t*-test corrected with the Benjamini-Hochberg procedure (FDR).

To further assess a potential direct role of HSPCs in angiogenesis, we performed a tube formation assay using HUVEC cultured in the presence and absence of HSPC-derived conditioned media. In this setting, we failed to detect a significant difference in the number of tubes or the average tube length between control or HSPC conditioned media. We provide these results below as **Fig. 2 for Reviewers only:**

Figure 2 for Reviewers only: HSPC conditioned media does not interfere with tube formation assay. HUVEC tube formation assay was performed with conditioned media of HSPCs or respective control media and cultured for 4 h before staining. Tube formation was imaged on a ZEISS ApoTome.2 Microscope (Zeiss). Average branch length

and total number of tubes were calculated from several fields in each sample using ImageJ software. ($n = 1$ experiment).

While tumor-associated HSPC may impact tumor angiogenesis, our experiments indicate contradicting results. We therefore cannot hypothesize with sufficient experimental evidence if tumor-associated HSPCs can impact on tumor angiogenesis. A more in-depth research effort e.g. by the analysis of in vivo models, may be required to address this aspect.

[5] *The CXCL12/CCR4 is a well know axis to recruit HSC, NSCs, T cells and cancer stem cells. Did the authors test other chemokine/ligand axes of potential migration to tumor? Please show this data.*

Response: Following this Reviewer's suggestion, we compared HSC high and low tumors for the expression of chemokines/receptors transcripts in addition to CXCL12/CCR4. We could detect significant increase in several chemokine/receptors axes such as:

- a) *IL-8* and its receptors *CXCR1*, *CXCR2*
- b) *CCL2* and *CCR2*, *CCR4*
- c) *CCL4* and *CCR5*
- d) *CCL5* and *CCR3*
- e) *CCL20* and *CCR6*.
- b) *CXCL12* and *CXCR4*

These data are presented in **Supplementary Fig. 11** of the revised manuscript.

In addition, we performed a cytokine array to measure 30 different cytokines, including 8 chemokines, in our patient-derived organoid co-cultures. In this experiment, we found a significant increase in chemokines IL-8, CCL2 and CCL4 secretion compared to tumor cell cultures or HSPCs alone (**Fig. 5j** of the revised paper). These findings indicated that additional chemokine/receptor axes known to exert a pro-tumorigenic role in glioblastoma^{12, 13, 14, 15} may be increased in brain tumors in the presence of HSPCs.

[6a]. *"The authors did an excellent job identifying stem and progenitor cell populations in tumor. However, they failed to address that HSPCs also differentiate into the immune cells found in the tumor including MDCs, TAMs, DCs. How do the HSPC*

numbers correlate with other myeloid derived progenitor populations known to differentiate from these cells”

Response: We thank the Reviewer for this positive comment. To correlate HSPC signals with other myeloid cell types detected by Syllogist, we generated a correlation matrix showing Pearson correlation coefficients and their significance between all HSPC subsets and mature immune cell subsets analyzed by Syllogist. HSPC subsets, especially the more committed progenitors of the myeloid lineages, correlated significantly with signals from mature myeloid cell types, e.g. monocytes and neutrophils. These results are now presented in **Supplementary Fig. 10** and referenced in the **Result section** of the manuscript.

[6b] *“...and were other myeloid cell populations more strongly correlated with outcomes than HSPCs?*

Response: We performed univariate Cox regression for all myeloid and lymphoid cell types resulting from Syllogist analysis and correlated these with clinical outcomes. Syllogist did detect weak associations with macrophages and neutrophil cell types in overall survival, however, these associations were not significant. The data are presented in **Supplementary Table 5** of the revised manuscript.

[7] *In the co-cultures where HSPC input was shown to enhance proliferation how stable in the Lin-HSPC phenotype during these cultures?*

Response: This is an interesting aspect we did not consider during initial submission. We performed flow cytometry of HSPCs in post co-culture experiments with patient-derived organoids. Interestingly, flow cytometry analysis of HSPCs after co-culture, revealed that patient-derived glioblastoma cells could effectively maintain subpopulations of CD38⁻ (immature) and CD38⁺ hematopoietic progenitors when compared to cultures without tumor cells, suggesting favourable conditions for the maintenance of HSPCs in glioblastoma. Additionally, within these organoids, we noticed the expansion of a CD45⁺CD34⁻ immune cell population, indicating that bone marrow CD34 positive cells may be actively differentiating in the presence of patient-derived glioblastoma cells. In summary, these results indicated that a) Glioblastoma

cells can promote HSPC maintenance (including CD38neg “early” progenitors) during at least 20 days in vitro and b) A subset of HSPCs differentiate to CD34-negative immune cells during this time span in culture. These results are now included in **Fig. 5i** of the revised manuscript.

[8] *Gliomas may drive differentiation of HSPCs to myeloid derived suppressor cells or other immature myeloid cells that may actually be contributing to tumor cell proliferation and not the HSPCs themselves. Do the investigators have evidence that HSPCs drive tumor growth or progeny derived from these cells?*

Response: To explore this interesting aspect raised by the Reviewer, we induced the differentiation of CD34 positive bone marrow-derived cells in vitro during 14 days, using the same protocol we have used for our colony forming assay (Fig. 3e,f). Upon confirming differentiation using CD45, CD34 and CD38 stainings, we co-incubated these cells with T98G cells for 48 hours and compared cell proliferation with co-culture experiments that were performed using the original CD34pos HSPC population. While we could observe a significant increase in the number of T98G cells in the presence of the original HSPCs, we failed to detect a significant increase in proliferation following incubation with the differentiated progenies, using the same experimental setting. The observed effect was very small, and these data are therefore shown below as **Fig. 3 for Reviewers only**. We tried as well to derive a population of differentiated cells from the organoid experiment, but we failed to obtain a pure population of immune cells in sufficient number to perform further experiments. While the role of myeloid cells, including MDSCs, is well known to promote glioblastoma progression^{16, 17}, at this stage we have not enough experimental evidence for a comparative analysis between HSPCs and their progenies in driving proliferation of glioblastoma cells. In our opinion, to address this complex question (together with the angiogenesis question raised in point 4b), we need additional sets of data (e.g. using in vivo models) that are more suited for an independent follow up manuscript. Unfortunately, this aspect remains currently unresolved and has been addressed in the **Discussion** as a limitation of the present study.

Figure 3 for Reviewer only: Comparison of HSPCs and their differentiated progenies in coculture with T98G cells.

a Flow cytometry profiles of living HSPCs and differentiated cells gated for CD45, CD34 and CD38. Top panels represent CD34+ purified bone marrow-derived cells (HSPCs). Bottom panels represent the differentiated progenies from the same HSPCs cultured in the presence of IL-3 (20 ng/mL), IL-6 (20 ng/mL), G-CSF (20 ng/mL), GM-CSF (20 ng/mL), SCF (50 ng/mL) for 14 days. **b** Stacked barplot of the indicated HSPCs and differentiated cells populations shown in **a**. **c** Co-culture of HSPCs or their differentiated progenies with T98G. Barplots represent percent increase of T98G cells compared to control either alone (white bars) or in the presence of HSPCs or differentiated cells (black bars). $n = 3$ technical replicates for each condition, p values computed by Student's t -test.

These are important and interesting findings that highlight the recruitment of HSPCs in the natural progression of gliomas and potential for contribution the glioma development.

Response: We thank the Reviewer for the encouraging comment and, despite the fact that we could not dissect in more detail the role of HSPCs versus their progenies during tumor progression, we hope to have addressed all other Reviewer's questions and concerns satisfactorily.

Reviewer #3

In the present manuscript by Lu et al., the authors investigate the immunologic networks that are present in brain tumors and reveal that hematopoietic stem/progenitor cells (HSPCs) are present in the tumor microenvironment and correlate with the disease progression. More specifically, the authors establish a reference-based deconvolution algorithm (Syllogist) and determined a signature for 43 different cell types (with a special focus on immune cell types). Subsequently, they profiled the cellular landscape of brain tumor tissues and, intriguingly, found increasing signals from HSPCs. They also verified the presence of cells with an HSPC phenotype (morphological and functional) and showed that direct contact between HSPCs and tumor cells is required for enhancing their proliferation and PD-L1 expression. Finally, they make a correlation between HSPCs' numbers and the disease progression. The authors have combined several different assays (bioinformatic, phenotypic, functional) to associate the presence of HSPCs with brain tumors progression. Overall, their findings are of great interest and importance with potential clinical applications. However, fundamental issues need to be addressed.

Response: We appreciate the Reviewer's precise comments, questions and helpful suggestions. We performed several new experiments, especially in comparing tumor-associated HSPCs with normal bone-marrow derived or circulating HSPCs from healthy individuals. We hope to have addressed the Reviewer's requests with adequate experiments described in the revised manuscript and below in a point by point response.

[1] *In Figure 1, the authors should be more specific of how each population was defined (especially HSCs and markers used for isolation).*

Response: We added information on how the various cell types were defined and isolated in the **Supplementary Table 6** of the revised manuscript.

[2] *In Figure 2: The authors present neural/glia as a common set of genes; however, they are known to have distinct signatures. The authors should add/split if one of them*

is more affected or in case they are both affected, they should comment on the clustering they did for the analysis.

Response: We agree with the Reviewer that the neural and glial cells have different ontogenies and split the data according to the Reviewer's request. We also split lymphoid cells in B and T cells respectively to achieve a more detailed view of the results that are presented in **Fig. 2b,d** of the revised manuscript.

[3] Moreover, based on Fig 2A, HSPCs seem to be expressed in healthy brains. The authors should comment more on whether HSC and/or HSPCs signature is found in the healthy brain (even if at much lower levels).

Response: We addressed this Reviewer comment by staining HPSCs in normal autaptic brain tissues. To do so, we stained for CD34 / CD45 on the inferior parietal lobule of the brain. When scanning through various anatomical brain regions, we were not able to detect HSPCs (**Fig. 4 for Reviewer's only** below). In contrast, using the same immunofluorescence protocol, we could detect double positive cells in 4/4 GBM patient specimen. (**Fig. 3a of the revised manuscript**)

Figure 4 for Reviewer only: Study of healthy inferior parietal lobule autopsy tissue. H&E (middle) and CD34 (red)/CD45 (green) dual immunofluorescence staining of the inferior parietal lobule of healthy autopsy brain. Nuclear counterstain of IF staining was performed using DAPI. Pictures were taken from representative regions (subpial, intracortical, grey/white matter and white matter). Scale bar = 100 μ m.

In addition to immunofluorescence, we conducted single cell sequencing of a CD45+CD34+ enriched GBM sample together with a control sample derived from a tumor-free region. While we could detect and sequence several HSPCs in the GBM

sample ($n = 105$ out of 783 cells sequenced), we failed to detect any HSPC from the tumor-free region sample ($n = 0$ out of 147 single cells). These results are shown in **Fig. 4e** of the revised manuscript.

In conclusion, we failed to detect HSPCs in “non-tumor” brain tissues. While further studies are warranted to characterize potential HSPCs in normal brains, our results point in the direction of a much rarer population (if any) compared to glioma tissues.

[4] *In Figure 3D: More cells appear to be CD34+ and CD45high (apart from the asterisk, which is an endothelial cell). Which cell type is represented with this? In general, the authors should provide more evidence on the cellular composition of the tumors examined (including lower magnification images).*

Response: To provide more evidence on the cellular composition of the GBM tissues, in regards to tumor-associated HSPCs, we performed additional immunofluorescence studies on a set of glioblastoma tissues and replaced the IF picture in Figure 3 with 2 other cases performed on formalin-fixed paraffin-embedded tissue sections. The new data is shown in **Fig. 3a** of the revised manuscript. We included below for the Reviewer only the Immunofluorescence staining included in the first version of the manuscript (right panel) with a lower a magnification image (left panel). These data, in contrast to the data in the revised main manuscript, were generated on a fresh frozen glioblastoma tissue.

Figure 5 for Reviewer only: Presence of CD34+ CD45+ cells in glioblastoma tissues. Overview and close-up immunofluorescence images of a CD34+CD45+ cell in fresh frozen glioblastoma tissue. Scale bars = 50 μm (overview, left) and 10 μm (cut out, right).

[5] Also, fundamental controls are missing: do healthy brains also contain HSCs? Less, more?

Response: In both immunofluorescence and single cell sequencing, we failed to detect HSPCs (including HSCs) in tumor-free regions, while these cells were readily found in GBM tissues. These results were also supported by the computational analysis shown in **Fig. 2a,b** and **Supplementary Fig. 2a**, where HSPC signals in normal brains are significantly less than in brain tumors. While we cannot demonstrate an absence of HSCs/HSPCs in healthy brain tissues, our results suggest a much lower (if any) presence of HSCs/HSPCs in healthy brains. We also refer to the previous response to point 3 for additional details on this aspect of the study.

[6] *In the same context, the authors state that tumor cells provide the necessary microenvironment for HSPC colonization (based on Fig. 3). One question rising is whether a higher number of HSPCs infiltrates the tissue or if they could proliferate more in the tumor microenvironment.*

Response: We believe that both, infiltration and proliferation, contribute to the higher number of tumor-associated HSPCs for the following reasons:

a) Infiltration: in several mouse models, ectopic injected HSPCs have been shown to readily and specifically infiltrate the tumor microenvironment. For this reason, HSPC carrying anti-cancer drugs have been proposed as a tool to access gliomas for therapeutic goals^{18, 19, 20, 21}. We have added in the **Discussion** a paragraph citing the most important studies covering preferential infiltration of HSPCs in gliomas.

b) Proliferation: In **Fig. 4** of the revised manuscript, we provide single cell analysis of tumor-associated HSPCs and noted a significant increase of cell cycle associated signatures in the more immature HSPC lineages such as HSCs and CMP, compared to bone marrow and circulating HSPCs. These data suggest that tumor-associated HSPCs may be expanding in situ (**Fig. 4f**).

[7] *Figure 3F is missing the quantification of CFUs derived from BM HSPCs.*

Response: We provide the quantification as requested in the updated **Fig. 3f**.

[8] Are HSPCs from brain tumors equally potent?

Response: We thank the Reviewer for this interesting question that has not been addressed during initial submission. To do so, we first looked for GBM samples with similar HSPC flow cytometry profiles, from which colony-forming data were available. We note for example, that patients 3, 8 and 14 have similar HSPC profiles by flow cytometry (**Fig. 3d** of the revised manuscript). By assuming equal potency, we would expect similar colony-forming activity *ex vivo*. However, under identical culture conditions, colonies from Patient 3, 8 and 14 produced distinct types of colonies (**Fig. 3f**), suggesting that HSPCs from brain tumors may have heterogeneous potency, depending on the patient. This finding has been additionally included as a short statement in the conclusion part of the result section of Figure 3.

[9] Also, further characterization from brain tumors HSPCs is required such as transcriptome compared to BM and brain healthy HSPCs, cell cycle analysis (Ki67 or Hoechst).

Response: To address the Reviewer's request on comparative transcriptome and cell cycle analysis of HSPCs, we performed single cell sequencing on control and tumor-derived HSPCs. We established a protocol for the isolation and enrichment of these cells and obtained the transcriptome derived from 104 single tumor-associated HSPCs that could be compared with healthy bone marrow and circulating HSPCs transcriptomes. These data are now shown in **Fig. 4** and **Supplementary Fig. 5** of the resubmitted manuscript. Briefly, by differential gene expression analysis, we could observe 3 genes (*APOE*, *HMOX1* and *SPP1*) consistently upregulated in tumor-associated HSPCs when compared to healthy bone marrow or circulating HSPCs (**Fig. 4g**). In particular, paracrine *SPP1* deriving from tumor associated immune cells, has been recently shown to mediate PTEN-dependent glioma progression²². Using the same dataset, we analyzed cell cycle-related gene expression signatures to differentiate cells in a cycling or non-cycling stage. For this purpose, we used the method introduced by Butler et al.²³, which leverages on the analysis of S-Phase and G2M-Phase cell cycle gene signatures (including *MKI67*), obtained using also HSPC as reference, among others²⁴. This method is commonly used to score cell cycle phases in single cell RNA-Seq experiments. Cell cycle profiles of our bone marrow-

derived HSPC dataset were similar to recent studies using the same scoring method^{25, 26}. To our surprise, we did not only detected cycling HSPCs in tumor-associated HSPCs, but the proportions of cycling HSPCs in glioblastoma were significantly higher compared to healthy bone marrow or circulating HSPCs (**Fig. 4f**) or compared to differentiated myeloid cells isolated from the same GBM sample (**Supplementary Fig. 5**). This data indicates that immature HSPC populations detected in brains (in particular HSCs and CMPs) may be actively proliferating in situ.

[10] Figure 4C: the authors should show FACS plots/gating strategy of HSPCs.

Response: We included the gating strategy in **Fig. 5a** of the revised manuscript

[11] *They also correlate HSC^{high} glioblastoma with increased TGF- β 1 and IL-10 (related with immunosuppression) and not pro-inflammatory cytokines. They should provide more evidence regarding the immunosuppressive effect of HSC^{high} in the tumor environment. The authors should address this by using the in vitro assay of co-cultures and measure the profile of cytokines (ELISA).*

Response: We appreciate the interesting suggestion of the Reviewer. We repeated our matrigel co-culture of patient-derived glioblastoma cells (patient 17) with and without HSPCs and subsequently performed a cytokine array to measure 30 different cytokines from the cell-free culture media, collected after 9 and 20 days of coincubation. Interestingly, after 20 days of co-culture, we detected a significant increase of known tumor promoting cytokines such as interleukin 6 (IL-6)²⁷ and IL-8^{12, 28} ($p = 7.5 \times 10^{-7}$ and $p = 0.0028$, respectively), or positive regulators of immunosuppression such as chemokine ligand 2²⁹ (CCL2, $p = 0.035$). In addition, we detected a significant increase of CCL4, ($p = 0.006$) and an increase of the soluble form of the tumor necrosis factor α receptor 1 (sTNF-R1) after 9 days in culture ($p = 0.004$). Other cytokines analyzed in our microarray were either not detected or, if detected, were not significantly different when compared with tumor cell cultures without HSPCs. These results are reported in **Figure 5j** and **Supplementary Fig. 8** of the revised manuscript. Additionally, when analyzing the TCGA dataset for HSC signals, we noted the upregulation of *IL6*, *IL8*, *CCL2* and *CCL4* gene expression in the

HSC-high tumors, in agreement with the cytokine microarray experiment. (**Supplementary Fig. 11**).

[12] In Figure 4E: They perform co-cultures of HSPCs with patient-derived organoids and show that they display increased organoid size. The authors should also perform a cytokine array.

Response: We performed the suggested cytokine array in the patient-derived organoids (see response to point [11]).

[13] In Figure 5: HSCs are usually a rare cell population. How do the authors explain such enrichment using brain tumors?

Response: HSPCs, including HSCs, can specifically infiltrate gliomas when injected intravenously in experimental models^{18, 19, 20, 21} (see also point 14 below as suggested by this Reviewer). Based on these works, we can hypothesize that circulating HSCs may preferentially migrate towards gliomas in vivo, resulting in specific enrichment within these tissues. In addition, we also observed that human glioblastomas expressed niche factors potentially enabling the maintenance of HSPCs within the tumor (Fig. 3g, h and Fig. 6h). These data were also supported by the maintenance of HSPCs subsets in the organoid experiment after 20 days, when compared with HSPC cultures alone (**Fig. 5i**). Finally, we observed a stronger enrichment of cell cycle-related genes in single tumor-associated HSPCs compared to circulating HSPCs or differentiated myeloid cells found in the same samples (**Fig. 4f** and **Supplementary Fig. 5b**). These data collectively imply that in humans, circulating HSPCs a) may preferentially migrate towards brain tumors, b) can be maintained in the tumor microenvironment, at least in vitro and c) may proliferate in situ. We have included a statement in the **Discussion**, that addresses this question by the Reviewer. We have also included references that describes the most important studies on HSPC migration to brain tumors (see also point 14 below).

[14] Discussion: It has been previously shown in animal models that HSPCs can infiltrate the brain tumor environment, further supporting the authors' findings. These

studies could also be implemented in the discussion (https://www.ncbi.nlm.nih.gov/pmc/articles/PMC5101999/).

Response: We included in the **Discussion** the paper kindly suggested by the Reviewer, together with references to other studies and a paragraph discussing their findings^{18, 19, 20, 21}.

[15] Material and Methods (line 609): HSC and MPPs: CD34+CD38+ (it should be corrected to CD38-)

Response: We notice the error and have corrected the sentence accordingly.

[16] Typo in line 500 (respons es)

Response: We notice the typing error and have corrected the sentence accordingly.

ADDITIONAL CHANGES INCLUDED IN THE REVISED MANUSCRIPT

During the course of manuscript revision, we updated the manuscript as follows:

a) We noticed inconsistencies of patient numbers in the Result section referring to Fig. 2c,d. The numbers have been updated to reflect the data presented in the corresponding Figure.

b) Statistical values in the revised Fig. 2b, d and Supplementary Fig. 2 were adjusted for multiple comparisons derived from additional Student's *t*-tests.

c) We noticed an error in the number of HSPCs per million stated for patient 14 and patient 12 above the stacked bar plot of our FACS data (original Fig. 3c, now Fig. 3d). We corrected the number of HSPCs per million from 1089 to 545 (patient 14) and from 3767 to 1884 (patient 12). Statistical evaluation of Fig. 3d (comparison of HSCs subpopulations between GBM and non-GBM tumors) was updated accordingly in the Result section.

e) One patient sample in Fig. 4f, g of the initial submission was mislabelled (Pat. 18). The correct labeling is Pat. 13 and was updated in the revised manuscript (now Fig. 5g and h).

f) In the Result section of Fig. 6 we now report the numbers of "GBM patients with clinical follow-up data available" ($n = 159$). In the initial submission we reported the wrong number that referred to the "GBM tissue samples available in the TCGA dataset" ($n = 172$).

g) Statistical values in the revised Fig. 6 were adjusted to reflect multiple testing derived from additional univariate Cox regression requested by Reviewer 2, point 6b. In the same Figure, the immunoscore statistics and aggregate measures of cell types were removed and substituted with more relevant statistical evaluation of all HSPC subsets (Fig. 6d of the revised manuscript). The requested additional evaluations are now part of a new supplementary table 5 in the revised manuscript). In addition, we provide a multivariable regression model in Fig. 6e to account for potential confounders, instead of the originally submitted independence tests. The multivariable Cox regression is the correct statistical method for assessing the effect of potential confounders in a survival analysis.

h) We have updated the author list and acknowledgements thereby including the people who contributed during revision.

REFERENCES

1. Aran D, Hu Z, Butte AJ. xCell: digitally portraying the tissue cellular heterogeneity landscape. *Genome Biol* **18**, 220 (2017).
2. Finotello F, *et al.* Molecular and pharmacological modulators of the tumor immune contexture revealed by deconvolution of RNA-seq data. *Genome Med* **11**, 34 (2019).
3. Newman AM, *et al.* Robust enumeration of cell subsets from tissue expression profiles. *Nat Methods* **12**, 453-457 (2015).
4. Racle J, de Jonge K, Baumgaertner P, Speiser DE, Gfeller D. Simultaneous enumeration of cancer and immune cell types from bulk tumor gene expression data. *Elife* **6**, (2017).
5. Schug J, Schuller WP, Kappen C, Salbaum JM, Bucan M, Stoeckert CJ, Jr. Promoter features related to tissue specificity as measured by Shannon entropy. *Genome Biol* **6**, R33 (2005).
6. Mabbott NA, Baillie JK, Brown H, Freeman TC, Hume DA. An expression atlas of human primary cells: inference of gene function from coexpression networks. *BMC Genomics* **14**, 632 (2013).
7. Cima I, *et al.* Tumor-derived circulating endothelial cell clusters in colorectal cancer. *Sci Transl Med* **8**, 345ra389 (2016).
8. Li H, *et al.* Reference component analysis of single-cell transcriptomes elucidates cellular heterogeneity in human colorectal tumors. *Nat Genet* **49**, 708-718 (2017).
9. Rozenblatt-Rosen O, Stubbington MJT, Regev A, Teichmann SA. The Human Cell Atlas: from vision to reality. *Nature* **550**, 451-453 (2017).
10. Walton NM, *et al.* Derivation and large-scale expansion of multipotent astroglial neural progenitors from adult human brain. *Development* **133**, 3671-3681 (2006).
11. Glas M, *et al.* Targeting the cytosolic innate immune receptors RIG-I and MDA5 effectively counteracts cancer cell heterogeneity in glioblastoma. *Stem Cells* **31**, 1064-1074 (2013).
12. Brat DJ, Bellail AC, Van Meir EG. The role of interleukin-8 and its receptors in gliomagenesis and tumoral angiogenesis. *Neuro Oncol* **7**, 122-133 (2005).
13. Vakilian A, Khorramdelazad H, Heidari P, Sheikh Rezaei Z, Hassanshahi G. CCL2/CCR2 signaling pathway in glioblastoma multiforme. *Neurochem Int* **103**, 1-7 (2017).

14. Wang Y, Liu T, Yang N, Xu S, Li X, Wang D. Hypoxia and macrophages promote glioblastoma invasion by the CCL4-CCR5 axis. *Oncol Rep* **36**, 3522-3528 (2016).
15. Infanger DW, *et al.* Glioblastoma stem cells are regulated by interleukin-8 signaling in a tumoral perivascular niche. *Cancer Res* **73**, 7079-7089 (2013).
16. Bayik D, *et al.* Myeloid-Derived Suppressor Cell Subsets Drive Glioblastoma Growth in a Sex-Specific Manner. *Cancer Discov* **10**, 1210-1225 (2020).
17. Hambardzumyan D, Gutmann DH, Kettenmann H. The role of microglia and macrophages in glioma maintenance and progression. *Nat Neurosci* **19**, 20-27 (2016).
18. Flores CT, *et al.* Lin(-)CCR2(+) hematopoietic stem and progenitor cells overcome resistance to PD-1 blockade. *Nat Commun* **9**, 4313 (2018).
19. Tabatabai G, *et al.* Lessons from the bone marrow: how malignant glioma cells attract adult haematopoietic progenitor cells. *Brain* **128**, 2200-2211 (2005).
20. Wildes TJ, *et al.* Cross-talk between T Cells and Hematopoietic Stem Cells during Adoptive Cellular Therapy for Malignant Glioma. *Clin Cancer Res* **24**, 3955-3966 (2018).
21. Bryukhovetskiy IS, *et al.* Hematopoietic stem cells as a tool for the treatment of glioblastoma multiforme. *Mol Med Rep* **14**, 4511-4520 (2016).
22. Chen P, *et al.* Symbiotic Macrophage-Glioma Cell Interactions Reveal Synthetic Lethality in PTEN-Null Glioma. *Cancer Cell* **35**, 868-884 e866 (2019).
23. Butler A, Hoffman P, Smibert P, Papalexi E, Satija R. Integrating single-cell transcriptomic data across different conditions, technologies, and species. *Nat Biotechnol* **36**, 411-420 (2018).
24. Tirosh I, *et al.* Single-cell RNA-seq supports a developmental hierarchy in human oligodendroglioma. *Nature* **539**, 309-313 (2016).
25. Bjorn N, Jakobsen I, Lotfi K, Green H. Single-Cell RNA Sequencing of Hematopoietic Stem and Progenitor Cells Treated with Gemcitabine and Carboplatin. *Genes (Basel)* **11**, (2020).
26. Canu G, *et al.* Analysis of endothelial-to-haematopoietic transition at the single cell level identifies cell cycle regulation as a driver of differentiation. *Genome Biol* **21**, 157 (2020).
27. Weissenberger J, *et al.* IL-6 is required for glioma development in a mouse model. *Oncogene* **23**, 3308-3316 (2004).

28. Sharma I, Singh A, Siraj F, Saxena S. IL-8/CXCR1/2 signalling promotes tumor cell proliferation, invasion and vascular mimicry in glioblastoma. *J Biomed Sci* **25**, 62 (2018).
29. Chang AL, *et al.* CCL2 Produced by the Glioma Microenvironment Is Essential for the Recruitment of Regulatory T Cells and Myeloid-Derived Suppressor Cells. *Cancer Res* **76**, 5671-5682 (2016).

REVIEWER COMMENTS

Reviewer #1 (Remarks to the Author):

The authors addressed all issues raised in the original review satisfactorily and thereby improved the manuscript considerably. Specifically, the authors carried out additional computational analyses (Fig 1c, d and Fig 1e) and performed additional measurements (Fig 3a, Fig 3c-f) to support their findings. Additionally, all other suggestions and corrections were included in the manuscript.

Reviewer #2 (Remarks to the Author):

The authors have submitted a significantly revised and improved manuscript that supports their observations of an association of HPSCs progenitors within glioma microenvironments as being associated with poorer prognosis and an immunosuppressive phenotype. Additional data and details regarding methods supports the major conclusions of the well-designed scientific report. The manuscript is responsive to prior critiques and will be of broad interest to the scientific community.

Reviewer #3 (Remarks to the Author):

One of the major questions that I had was regarding the presence of HSPCs in healthy brains. The authors have now addressed this (via immunofluorescence -for CD34 and CD45- in normal autoptic brain tissues and single cell sequencing of a CD45+CD34+ enriched GBM sample together with a control sample derived from a tumor-free region) and found that they could not detect HSPCs in “non-tumor” brain tissue. Additionally, the authors performed single cell sequencing on control (BM or circulating) and tumor-derived HSPCs and analyzed cell cycle-related gene expression signatures to differentiate cells in a cycling or non-cycling stage and found that they are more proliferative in the niche – which was also one of the main questions of this reviewer. Lastly, the authors have performed the cytokine array that was asked and identified a few cytokines significantly enriched (tumor promoting/immunosuppressive) in the presence of HSPCs.

To sum up, the authors have significantly enriched this study and all my former comments have been carefully addressed.

Reviewer #4 (Remarks to the Author):

In their study, Lu and colleagues report the presence of HSPCs in glioblastoma, and a link between tumor-associated HSPCs and immunosuppression. The manuscript is clearly of high interest. I was asked to comment on the single-cell RNA-seq dataset that the authors have added to further characterise the tumor-associated HSPCs.

Single-cell RNA-seq of CD45+CD34+ cells from GBM was used to confirm that GBM-associated hematopoietic/immune cells are immature HSPCs (i.e. HSCs or MPPs). Whether or not this is the case is currently difficult to assess given the information presented by the authors.

a) Cell type annotation is entirely done by sylogist. The authors do not validate the use of this novel tool for the automated annotation of single cell RNA-seq datasets; validation is only shown for cell type decomposition of bulk RNA datasets. Standard tools should be applied, for example projection on reference cell atlases (e.g using the algorithms from [10.1016/j.cell.2019.05.031](https://doi.org/10.1016/j.cell.2019.05.031) or <https://cblast.gao-lab.org> and the datasets from [10.1038/ncb3493](https://doi.org/10.1038/ncb3493), [10.1038/s41467-019-10291-0](https://doi.org/10.1038/s41467-019-10291-0) or [10.1038/s41586-020-2157-4](https://doi.org/10.1038/s41586-020-2157-4) as reference.)

b) Besides automated annotation, manual analysis of marker genes is useful for identifying HSPC

subtypes. The amount of information currently shown in the manuscript is insufficient to evaluate the annotation. Minimally the authors should (i) include lists of genes overexpressed by each cluster from fig. 4b in the supplement, (ii) show t-SNE plots highlighting the expression of the markers commonly used in the field to classify human HSPC subsets, see for example 10.1038/ncb3493; (iii), for the same set of markers, display boxplots comparing their expressing in the different putative HSPC subsets shown in figure 4e, i.e. putative HSCs, CMPs, GMPs etc.; and (iv) include lists of genes overexpressed by each of these groups in the supplement.

c) In figure 4f, GBM-HSPCs and Bone-marrow/blood-HSPCs cluster clearly apart yet only 4 genes were identified to be significantly differentially expressed. This is very unusual. I would ask the authors to provide the full result of this comparison (p values, log fold changes for all genes in the genome) in the supplement and specify the statistical test used. Ref. 76 is cited in this context but does not specify a statistical test. MAST (10.1186/s13059-015-0844-5) is a suitable test that is implemented in the Seurat package.

d) The GEO dataset (GSE165238) does not include relevant metadata (in particular, whether each single cell is from glioblastoma, tumor-free brain, bone marrow or PBMC; but also, the coordinates of each cell in the t-SNEs shown in the manuscript, and which cluster the cell belongs to). Having such metadata available would make it much easier to assess the cell type classification presented in the manuscript.

A careful annotation of the single cell data is particularly important since the FACS data shown in figure 3c,d is in my opinion not conclusive. The FACS panel used by the authors identifies immature HSPCs exclusively on the basis of absence of staining for immune lineage markers, CD34 expression, and absence of CD38 expression. HSCs are additionally identified by CD90 expression and absence of CD45RA expression. However, the only positive markers used in this scheme (CD34 and CD90) are expressed by non-hematopoietic cells, such as mesenchymal and endothelial cells (see datasets cited below). This scheme is therefore commonly used in bone marrow, where non-hematopoietic cells are rare. In other tissues such cells may be much more frequent. The authors claim that 'the presence of non-hematopoietic progenitors expressing CD34 such as endothelia and tumor cells was excluded from (the) analysis by the lineage markers CD14 and CD56 respectively'. This statement may be incorrect.

a) CD14 is not expressed by primary endothelial cells: See <http://betsholtzlab.org/VascularSingleCells/database.html> for a dataset on mouse brain vasculature and <https://db.cngb.org/HCL/landscape.html> for a human cell atlas. The reference cited by the authors found CD14 expressed on the HUVEC cell line, which cannot be considered representative of primary endothelial cells.

b) Even leaving endothelial cells aside (the authors provide an additional control by adding anti-CD144 to the lineage cocktail), CD34 and Thy1 (CD90) are also expressed on many mesenchymal/fibroblast cell types(see cell atlas resources cited in previous paragraph), and possibly NSC subsets. The current gating strategy does not exclude these cells.

Taken together I am skeptical of the experiments shown in figure 3c,d. A possible solutions would be to pre-gate with CD45+ (to only include hematopoietic/immune cells); personally I also would consider a more detailed annotation of the scRNA-seq sufficient to support the statements on immature phenotypes of GMB-associated HSPCs. In that case I would recommend that the FACS data should be excluded.

Lu et al. Tumor-associated hematopoietic stem and progenitor cells positively linked to glioblastoma progression

POINT BY POINT RESPONSE TO REVIEWERS

We would like to thank Reviewer 4 for the constructive comments and suggestions and Reviewers 1-3 for the positive feedback to our revised manuscript. All changes to the second revision of this manuscript are reported here and highlighted in the accompanying manuscript in yellow color. References cited in this response to Reviewers are listed at the end of this document.

Reviewer #1

General comments

The authors addressed all issues raised in the original review satisfactorily and thereby improved the manuscript considerably. Specifically, the authors carried out additional computational analyses (Fig 1c, d and Fig 1e) and performed additional measurements (Fig 3a, Fig 3c-f) to support their findings. Additionally, all other suggestions and corrections were included in the manuscript.

Response: We thank this Reviewer for the positive feedback and for the suggested experiments that helped us to improve the manuscript.

Reviewer #2

General comments

The authors have submitted a significantly revised and improved manuscript that supports their observations of an association of HPSCs progenitors within glioma microenvironments as being associated with poorer prognosis and an immunosuppressive phenotype. Additional data and details regarding methods supports the major conclusions of the well-designed scientific report. The manuscript is responsive to prior critiques and will be of broad interest to the scientific community.

Response: We thank this Reviewer for the positive feedback. We are grateful for comments and suggestions for experiments related to HSPCs, which helped us to improve the quality of our manuscript.

Reviewer #3

General comments

One of the major questions that I had was regarding the presence of HSPCs in healthy brains. The authors have now addressed this (via immunofluorescence -for CD34 and CD45- in normal autaptic brain tissues and single cell sequencing of a CD45+CD34+ enriched GBM sample together with a control sample derived from a tumor-free region) and found that they could not detect HSPCs in “non-tumor” brain tissue. Additionally, the authors performed single cell sequencing on control (BM or circulating) and tumor-derived HSPCs and analyzed cell cycle-related gene expression signatures to differentiate cells in a cycling or non-cycling stage and found that they are more proliferative in the niche – which was also one of the main questions of this reviewer. Lastly, the authors have performed the cytokine array that was asked and identified a few cytokines significantly enriched (tumor promoting/immunosuppressive) in the presence of HSPCs.

To sum up, the authors have significantly enriched this study and all my former comments have been carefully addressed.

Response: We appreciate the throughout positive feedback of this Reviewer considering our revised manuscript. We are grateful for the very pertinent questions which prompted us to generate the single cell RNA-Seq dataset.

Reviewer #4

General comments

In their study, Lu and colleagues report the presence of HSPCs in glioblastoma, and a link between tumor-associated HSPCs and immunosuppression. The manuscript is clearly of high interest. I was asked to comment on the single-cell RNA-seq dataset that the authors have added to further characterise the tumor-associated HSPCs. Single-cell RNA-seq of CD45+CD34+ cells from GBM was used to confirm that GBM-associated hematopoietic/immune cells are immature HSPCs (i.e. HSCs or MPPs). Whether or not this is the case is currently difficult to assess given the information presented by the authors.

Response: We thank the Reviewer for thoroughly commenting our single cell and FACS data. We are also grateful for the references and methods suggested by the Reviewer that helped us to address the concerns raised. A point-by-point response is included below.

[1] *Cell type annotation is entirely done by syllogist. The authors do not validate the use of this novel tool for the automated annotation of single cell RNA-seq datasets; validation is only shown for cell type decomposition of bulk RNA datasets. Standard tools should be applied, for example projection on reference cell atlases (e.g using the algorithms from 10.1016/j.cell.2019.05.031 or <https://cblast.gao-lab.org> and the datasets from 10.1038/ncb3493, 10.1038/s41467-019-10291-0 or 10.1038/s41586-020-2157-4 as reference.)*

Response: Syllogist was indeed validated on bulk RNA-Seq data only. Following the Reviewer's suggestion, we now annotated our datasets with a previously validated algorithm. To this end we explored several tools, including the ones kindly suggested by the Reviewer. We decided to annotate our datasets using SingleR and a reference based on data provided by ENCODE and Blueprint datasets (259 RNA-seq samples of pure stroma and immune cells)¹. This reference includes, among others, neural and hematopoietic stem cell types in a single normalized data set and has been used in the referenced publication. For consistency, all annotations of single cell datasets were

performed using this method and reference. The newly annotated data are included in the revised manuscript as **Fig. 4 and Supplementary Fig. 5**.

[2] Besides automated annotation, manual analysis of marker genes is useful for identifying HSPC subtypes. The amount of information currently shown in the manuscript is insufficient to evaluate the annotation. Minimally the authors should (i) include lists of genes overexpressed by each cluster from fig. 4b in the supplement.

Response: We included the requested information in **Supplementary Fig. 5a** as a heatmap showing the top differentially regulated genes using the Seurat package. Markers for cluster 3 (containing HSPCs annotated by SingleR) contained genes previously shown to be overexpressed by hematopoietic progenitors (e.g., *SPINK2*).

[3] show t-SNE plots highlighting the expression of the markers commonly used in the field to classify human HSPC subsets, see for example 10.1038/ncb3493;

Response: We included in the **Supplementary Fig. 5c** the UMAP plots highlighting the expression of markers previously used to differentiate between human bone marrow-derived HSPC subsets². In this figure, we note that the expressed genes were located in separate regions of the HSPC cluster, matching with the regions occupied by the annotation of our bone marrow single cells. Moreover, the annotated HSPC subsets from the glioblastoma sample matched these regions, confirming similarity of the tumor-derived HSPC subsets with canonical subsets (except for the cells annotated as CLP, which were subsequently excluded from further analyses). We thank the Reviewer for this suggestion, which enabled us to visualize the similarity between bone marrow resident and tumor-derived HSPCs.

[4] for the same set of markers, display boxplots comparing their expressing in the different putative HSPC subsets shown in figure 4e, i.e. putative HSCs, CMPs, GMPs etc

Response: This information is shown in **Supplementary Fig. 5d**.

[5] include lists of genes overexpressed by each of these groups in the supplement.

Response: We included the information in **Supplementary Fig. 5g**. In the Bone marrow sample, we could identify several genes that were significantly regulated between subsets, among them genes matching the markers reported by Pellin *et al.*². This data represents further validation for the HSPC annotation using SingleR. In the glioblastoma dataset, only fewer genes could distinguish significantly HSPC subsets, among them *MPO*, *ELANE* and *AZU1*, three markers for the GMP subset in Pellin *et al.*². This may be due to the low number of cells included in the analysis. The top overexpressed genes for each subsets are presented as heatmaps in **Supplementary Fig. 5g** with a note (asterisk) denoting the significantly overexpressed genes.

[6] In figure 4f, GBM-HSPCs and Bone-marrow/blood-HSPCs cluster clearly apart yet only 4 genes were identified to be significantly differentially expressed. This is very unusual.

Response: We reanalyzed the aggregated data and performed integration using the Seurat package instead of the AGG function of Cellranger as previously submitted. After preprocessing and normalization, we observed that annotated HSPCs clustered in a common region, independently from the sample type. This is different from the previous analysis, and we conclude that the Seurat algorithm is a better normalization and integration approach compared to the one we used previously. We therefore used the resulting normalized data computed from Seurat for all follow-up analyses.

[7] I would ask the authors to provide the full result of this comparison (p values, log fold changes for all genes in the genome) in the supplement and specify the statistical test used. Ref. 76 is cited in this context but does not specify a statistical test. MAST (10.1186/s13059-015-0844-5) is a suitable test that is implemented in the Seurat package.

For differential gene expression analysis, we originally used the SSeq test implemented in the Cellranger package³. However, we followed this Reviewer's advice and performed DGE analysis using the MAST test, also in conjunction with a recent article indicating MAST as one of the better DGE tests for scRNA Seq⁴. All significantly

regulated genes and *p* values are now reported in **Supplementary Table 5-8**, with a selection of top regulated genes for each subset shown in **Fig. 4h**.

[8] *The GEO dataset (GSE165238) does not include relevant metadata (in particular, whether each single cell is from glioblastoma, tumor-free brain, bone marrow or PBMC; but also, the coordinates of each cell in the t-SNEs shown in the manuscript, and which cluster the cell belongs to). Having such metadata available would make it much easier to assess the cell type classification presented in the manuscript.*

Response: The meta data is available for each sample of the GEO dataset and can be accessed by following the sample link. However, to facilitate the accession of the meta data for follow-up analyses, we will provide the respective meta data in a separate sheet in GEO. The RNA-Seq data are now available under the following **GEO Accession: GSE165238 using the password (token): kvyduouyblelfcf**

[9] *A careful annotation of the single cell data is particularly important since the FACS data shown in figure 3c,d is in my opinion not conclusive.*

a) The FACS panel used by the authors identifies immature HSPCs exclusively on the basis of absence of staining for immune lineage markers, CD34 expression, and absence of CD38 expression. HSCs are additionally identified by CD90 expression and absence of CD45RA expression. However, the only positive markers used in this scheme (CD34 and CD90) are expressed by non-hematopoietic cells, such as mesenchymal and endothelial cells (see datasets cited below). This scheme is therefore commonly used in bone marrow, where non-hematopoietic cells are rare. In other tissues such cells may be much more frequent. The authors claim that ‘the presence of non-hematopoietic progenitors expressing CD34 such as endothelia and tumor cells was excluded from (the) analysis by the lineage markers CD14 and CD56 respectively’. This statement may be incorrect. CD14 is not expressed by primary endothelial cells: See <http://betsholtzlab.org/VascularSingleCells/database.html> for a dataset on mouse brain vasculature and <https://db.cngb.org/HCL/landscape.html> for a human cell atlas. The reference cited by the authors found CD14 expressed on the HUVEC cell line, which cannot be considered representative of primary endothelial cells. Even leaving endothelial cells aside (the authors provide an additional control by adding anti-CD144 to the lineage cocktail), CD34 and Thy1 (CD90) are also expressed

on many mesenchymal/fibroblast cell types(see cell atlas resources cited in previous paragraph), and possibly NSC subsets. The current gating strategy does not exclude these cells. Taken together I am skeptical of the experiments shown in figure 3c,d. A possible solutions would be to pre-gate with CD45+ (to only include hematopoietic/immune cells); personally I also would consider a more detailed annotation of the scRNA-seq sufficient to support the statements on immature phenotypes of GMB-associated HSPCs. In that case I would recommend that the FACS data should be excluded.

Response: The Reviewer's concern about the potential contaminations of non-hematopoietic cells in the flow cytometric data is well-founded. CD34 positive cells of non-hematopoietic origin, particularly endothelial cells, may be falsely gated and defined as immature hematopoietic progenitors. Our intention has been to use a gold standard method applied over several years for the profiling of bone marrow HSPCs, which also contains abundant endothelial and stromal cells. We have followed the Reviewer suggestion to pre-gate for CD45 positive cells on a new set of 4 glioblastoma patients (baseline characteristics listed in **Supplementary Table 4**), in order to exclude potential contaminants that may misrepresent the proportions of HSPC subsets in brain tumors. In this experiment, now reported in **Supplementary Fig. 4c**, we indeed noted a reduction of absolute numbers of HSPC subsets in the tumor and blood samples when gating for CD45 positive cells. This may be due to the exclusion of non-hematopoietic cells as suggested by this Reviewer, but may also be a result of the exclusion of CD45 negative hematopoietic progenitors⁵. Despite this reduction, we also noted that the proportions of early CD38 negative (HSC and MPP) and late CD38 positive cells were comparable in the CD45-gated and non-gated samples. Both gating strategies showed an enrichment of early progenitors compared to the positive controls. In light of this control experiment, we believe that the data shown in Figure 3d can be considered valid, as gating for CD45, while reducing the number of HSPCs in some samples, did not change the proportion of early to late HSPC subsets (Barplot of **Supplementary Fig. 4c**). To independently test this claim, we also annotated a publicly available single cell RNA-Seq dataset comprising of 9 glioblastoma samples from 7 patients⁶ and compared it with annotations from 2 bone marrow samples⁷. In this dataset, HSPCs could be detected in all glioblastoma samples. In 7/9 samples the

proportion of early progenitors (HSCs and MPPs) in the HSPC annotated cells ranged from 10.5 to 76.8 %. In the bone marrow samples, the early progenitors ranged between 0.27 to 0.55 %. This data is now included in **Supplementary Fig. 4d** and supports our FACS data. We thank the Reviewer for this critical observation on the flow cytometry experiment and his suggestion on how to control for potential contaminants, which prompted us to perform additional control experiments.

Concerning the lineage marker expression of potential contaminating cells, we have updated the references to include better evidence on the expression of CD14, CD56 and CD38 on endothelial cells and absence of CD34 on mesenchymal stromal cells of the brain.

Brain endothelial cells, which are mostly CD34 positive, may also express the following lineage markers: CD38 (this would exclude them from the early progenitor subsets)⁸, CD56 (NCAM)⁹ and CD14¹⁰. Concerning stromal cells of mesenchymal origin, we are confident that the vast majority of these cells in the brain do not express CD34 and are therefore excluded from our analysis. This was previously shown on flow cytometric profiles of mesenchymal-like stem cells derived from 26 glioma patients¹¹ (Table 1). Mesenchymal stromal cells, at least in the bone marrow, do also express CD56¹². Other cells of neural origin, such as NSCs are also excluded by CD56 antibodies as they express this marker abundantly¹³. Of note, the anti-CD56 antibody used in this study (clone NCAM16.2) recognizes the polysialylated form of NCAM as well (PSA-NCAM¹⁴). In summary, while we agree that potential lineage negative, CD34 positive cells of non-hematopoietic origin may be detected using this FACS approach on brain tissues, we think that our major claim derived from this experiment (being the overrepresented presence of immature HSPC subsets in brain compared to bone marrow) still holds true, also in light of the new CD45 pre-gating experiment shown in **Supplementary Fig. 4c** and the independent annotation of a publicly available scRNA-Seq dataset of glioblastoma patients (**Supplementary Fig. 4d**).

REFERENCES

1. Aran D, *et al.* Reference-based analysis of lung single-cell sequencing reveals a transitional profibrotic macrophage. *Nat Immunol* **20**, 163-172 (2019).
2. Pellin D, *et al.* A comprehensive single cell transcriptional landscape of human hematopoietic progenitors. *Nat Commun* **10**, 2395 (2019).

3. Yu D, Huber W, Vitek O. Shrinkage estimation of dispersion in Negative Binomial models for RNA-seq experiments with small sample size. *Bioinformatics* **29**, 1275-1282 (2013).
4. Sonesson C, Robinson MD. Bias, robustness and scalability in single-cell differential expression analysis. *Nat Methods* **15**, 255-261 (2018).
5. Boulais PE, *et al.* The Majority of CD45(-) Ter119(-) CD31(-) Bone Marrow Cell Fraction Is of Hematopoietic Origin and Contains Erythroid and Lymphoid Progenitors. *Immunity* **49**, 627-639 e626 (2018).
6. Neftel C, *et al.* An Integrative Model of Cellular States, Plasticity, and Genetics for Glioblastoma. *Cell* **178**, 835-849 e821 (2019).
7. Han X, *et al.* Construction of a human cell landscape at single-cell level. *Nature* **581**, 303-309 (2020).
8. Boslett J, Hemann C, Christofi FL, Zweier JL. Characterization of CD38 in the major cell types of the heart: endothelial cells highly express CD38 with activation by hypoxia-reoxygenation triggering NAD(P)H depletion. *Am J Physiol Cell Physiol* **314**, C297-C309 (2018).
9. Gingras MC, Roussel E, Bruner JM, Branch CD, Moser RP. Comparison of cell adhesion molecule expression between glioblastoma multiforme and autologous normal brain tissue. *J Neuroimmunol* **57**, 143-153 (1995).
10. Jersmann HP. Time to abandon dogma: CD14 is expressed by non-myeloid lineage cells. *Immunol Cell Biol* **83**, 462-467 (2005).
11. Hossain A, *et al.* Mesenchymal Stem Cells Isolated From Human Gliomas Increase Proliferation and Maintain Stemness of Glioma Stem Cells Through the IL-6/gp130/STAT3 Pathway. *Stem Cells* **33**, 2400-2415 (2015).
12. Skog MS, *et al.* Expression of neural cell adhesion molecule and polysialic acid in human bone marrow-derived mesenchymal stromal cells. *Stem Cell Res Ther* **7**, 113 (2016).
13. Bonfanti L. PSA-NCAM in mammalian structural plasticity and neurogenesis. *Prog Neurobiol* **80**, 129-164 (2006).
14. Moebius JM, Widera D, Schmitz J, Kaltschmidt C, Piechaczek C. Impact of polysialylated CD56 on natural killer cell cytotoxicity. *BMC Immunol* **8**, 13 (2007).

REVIEWER COMMENTS

Reviewer #4 (Remarks to the Author):

The authors have substantially improved their single-cell transcriptomic and flow cytometric analyses. In my view, the new data and analyses provide clear evidence for the statements drawn in the manuscript and I can recommend its publication.

One final suggestion I would like to make would be to rephrase the statement in the abstract "In contrast to the medullary hematopoietic compartment, tumor-associated HSPCs are largely comprised of CD38- immature cells" to "Compared to the medullary hematopoietic compartment, tumor-associated HSPCs contain a higher fraction of immunophenotypically and transcriptomically immature, CD38- cells". This phrasing would more accurately reflect the nature of the data and the quantitative results.

Lu, Dobersalske *et al.* Tumor-associated hematopoietic stem and progenitor cells positively linked to glioblastoma progression.

POINT BY POINT RESPONSE TO REVIEWERS

Reviewer#4

The authors have substantially improved their single-cell transcriptomic and flow cytometric analyses. In my view, the new data and analyses provide clear evidence for the statements drawn in the manuscript and I can recommend its publication.

One final suggestion I would like to make would be to rephrase the statement in the abstract "In contrast to the medullary hematopoietic compartment, tumor-associated HSPCs are largely comprised of CD38- immature cells" to "Compared to the medullary hematopoietic compartment, tumor-associated HSPCs contain a higher fraction of immunophenotypically and transcriptomically immature, CD38- cells". This phrasing would more accurately reflect the nature of the data and the quantitative results.

Response: We thank the Reviewer for assessing our revised manuscript and for the positive feedback. We have rephrased the statement in the abstract as suggested.